# Laccase-Catalyzed Oxidation of Mixed Aqueous Phenolic Substrates at Low Concentrations

**Stoyan Rangelov and Jim A. Nicell \***

Department of Civil Engineering & Applied Mechanics, McGill University, 817 Sherbrooke Street West, Montreal, QC H3A 0C3, Canada; stoyanran@yahoo.com

**\*** Correspondence: jim.nicell@mcgill.ca; Tel.: +1-(514)-398-6675

**Abstract:** It has been proposed that *Trametes versicolor* laccase can be used to detoxify wastewaters that are contaminated with phenolic pollutants. However, the oxidation of phenols at low concentrations may be impacted if other substrates tend to interfere with or enhance the oxidation of the target substrate. To test this, experiments were conducted to evaluate effects arising from the simultaneous presence of mixed substrates including phenol (P), estradiol (E2), cumylphenol (CP), and triclosan (TCL), each of which are characterized by different rates of oxidation and tendencies to inactivate laccase. Slower and faster substrates were found to have only minor negative impacts upon the rate of conversion of targeted substrates, except where they tended to cause inactivation. No enhancements in substrate oxidation were observed. A multi-substrate kinetic model was shown to be able to accurately predict the time course of reactions of mixed substrates over extended periods at micromolar and sub-micromolar concentrations, except when estradiol and triclosan were simultaneously present. In this case, more enzyme inactivation was observed than would be expected from the oxidation of individual substrates alone. The utility of the model for providing insights into the reaction phenomenon and for evaluating the feasibility of oxidizing targeted substrates in the presence of other substrates is demonstrated.

**Keywords:** biocatalysis; enzymes; *Trametes versicolor* laccase; multi-substrate; oxidation; phenols; kinetic model

---

## 1. Introduction

Numerous studies have reported on the use of laccase to oxidize various phenols of environmental concern and, consequently, many assertions have been made that the enzyme has the potential to be used to eliminate these pollutants from wastewaters [1–4]. In these studies, experiments were conducted using initial substrate concentrations that were typically orders of magnitude greater than those encountered in real wastewaters. Moreover, excessive quantities of enzyme were used to accomplish significant substrate removal within reasonable time periods. In addition, authors failed to compare the final concentrations of substrates achieved through oxidation to regulatory discharge limits or other environmentally relevant criteria. As such, the practical feasibility of this application of laccase remains in question.

One approach to evaluating the feasibility of using enzymes to oxidize various contaminants at realistic wastewater concentrations was reported recently [5,6]. Specifically, a model was developed that accurately predicted the transient kinetics of the laccase-catalyzed oxidation of aqueous phenolic substrates at concentrations in the micromolar to sub-micromolar range [5,6]. This model was designed to predict the time course of the enzymatic oxidation of a substrate while accounting for the effects of unproductive side reactions and enzyme inactivation. When the model was applied to four different substrates, including phenol (P), 4-cumylphenol (CP), triclosan (TCL), and estradiol (E2), predictions

of substrate concentration as a function of time proved to be very accurate over several orders of magnitude of substrate and laccase concentrations and over prolonged reaction periods. The model was then used to demonstrate that substantial conversion (i.e., 90% to 99.9%) of estradiol, cumylphenol, and triclosan could be achieved for initial substrate concentrations in the micromolar and sub-micromolar range using pragmatic reaction times and laccase concentrations [5,6]. This is in contrast to phenol, a slow substrate of laccase, which was shown not to be practically amenable to substantial oxidation in the same concentration range [5]. The utility of the model was demonstrated by showing that it could be used to estimate the quantities of enzyme and the reaction times required to achieve target levels of conversion of individual substrates. Furthermore, the model could serve as a basis for reactor design and optimization, and for estimating operating and capital costs for reactor systems.

An important drawback of this model, however, is that it does not account for the possibility that more than one substrate may be present in a reacting mixture. This could be of particular importance for applications where one may wish to selectively target the oxidation of certain aqueous pollutants that, due to their particularly problematic nature, represent important hazards even when they are present at very low concentrations [5,6]. For example, even when in nanomolar concentrations in surface waters, the estrogenic properties of estradiol can have important impacts on fish species [7,8]. In instances where more than one phenolic substance may be present in a reaction matrix, such as in wastewaters containing multiple phenols in varying quantities and with different environmental impacts, it will be important to understand the effects of multiple substrates on each other's rate of conversion and on enzyme activity.

Mixture effects will likely depend on the affinity of the enzyme for the different substrates as reflected by their respective rates of oxidation, their tendencies to inactivate the enzyme, and the relative quantities of each of the substrates in the mixture. For instance, the rate of oxidation of a target substrate that is present in a mixture of other substrates, which are termed "secondary substrates" in this study, could conceivably be influenced by several factors. First, the rate of oxidation of the target substrate could be hindered by competition between it and secondary substrates of the enzyme (i.e., where the secondary substrates could occupy the active site during the catalytic process), thereby limiting access of the target substrate to the enzyme. For instance, it is hypothesized that slower secondary substrates that might tend to occupy the active site for longer periods might have a greater negative impact on the rate of oxidation of the target substrate than faster secondary substrates. Alternatively, a fast secondary substrate could be preferentially oxidized by the enzyme, resulting in a sequential delay in the oxidation of the slower target substrate until most of the secondary substrate is depleted. Secondly, given that different substrates can cause very different rates of inactivation of the enzyme [5,6], the presence of a secondary substrate that inactivates laccase during its oxidation could gradually cause a decrease in the rate of oxidation of the target substrate over time. Inactivation during the reaction of mixed substrates could conceivably be manifested in an additive manner, where the observed rate of inactivation would simply be the sum of the rates of inactivation arising from the oxidation of individual substrates. Alternatively, given that the free-radical products arising from the enzyme-catalyzed oxidation of mixed substrates can combine to form dimers, oligomers, or polymers covalently coupled by C–C, C–O and C–N bonds [9,10], these products could have differing tendencies to inactivate or inhibit the enzyme than those typically produced in single-substrate reactions. Thirdly, such polymeric products could themselves act as secondary substrates of laccase [11,12], and could tend to influence the rate of oxidation of the target substrate. Fourthly, an enhancement in the rate of oxidation of the target substrate could arise from the oxidation of secondary substrates and/or mixed reaction products where the free radicals produced could, in turn, attack and oxidize the target substrate [10]. An understanding of such interactions will be important for any application of laccase where multiple substrates may be present.

The objective of the present study is to expand the current understanding of enzyme kinetics by investigating the effects of the presence of secondary substrates on the laccase-catalyzed oxidation of a targeted substrate at concentrations in the micromolar and sub-micromolar range. The findings

will then be used to adapt the kinetic model reported previously, to expand its ability to model the time-course of reactions of multiple substrates in reacting mixtures. Finally, the insights gained from this work will be applied in the context of treating problematic phenolic pollutants to assess the feasibility of using laccase to catalyze the oxidation of micromolar and sub-micromolar concentrations of selected substrates in mixtures with other substrates.

Laccase from *Trametes versicolor* was chosen as the biocatalyst, since it has wide substrate specificity, has optimal activity in the acidic range but can function well at or around neutral pH (which is important for wastewater applications), and is catalytically stable [13,14]. Four substrates of laccase were selected for the purposes of this study, namely, 17β-estradiol (estradiol), 4-cumylphenol (cumylphenol), triclosan, and phenol. Each of these substrates represent diverse groups of contaminants of environmental importance and were the subjects of earlier studies of laccase-catalyzed oxidation where, as described above, their oxidation kinetics were successfully modeled over a wide range of substrate and enzyme concentrations [5,6].

## 2. Results and Discussion

### 2.1. Effects of Secondary Substrates on Rate of Oxidation of Target Substrates

Experiments were designed to determine whether one or more of the hypothesized effects described above would tend to occur in reactions of mixed substrates. Parallel 3-hour (h) batch reactions were conducted (each in triplicate) with single and mixed substrates. Mixed reactions were conducted over a range of concentrations of target and secondary substrates. Given that it was of particular interest to observe the effects of slow substrates on the rate of oxidation of fast substrates and vice versa, the four substrates were first classified in terms of their relative rates of oxidation. This was done using the single-substrate model presented in Appendix A in conjunction with the rate classification approach described in Appendix B. From this, estradiol was classified as "fast", cumylphenol and triclosan were classified as "intermediate", and phenol was classified as "slow". Similarly, as is also shown in Appendix B, the substrates were classified in terms of their tendency to inactivate the enzyme, with cumylphenol being classified as "high", estradiol and triclosan as "moderate", and phenol as "non-inactivating". Note that only estradiol, cumylphenol, and phenol were used in these experiments, since these three substrates were sufficient to compare the effects of compounds in terms of their relative rates of oxidation and tendencies to cause enzyme inactivation. The fourth substrate, triclosan, was introduced into subsequent experiments, as will be described in Section 2.2.2 below.

2.1.1. Effects of Slow Secondary Substrates on Fast Target Substrates

The effects of slow secondary substrates on the rate of oxidation of faster target substrates were investigated under three scenarios, as follows: (1) effect of phenol (slow rate, non-inactivating) on the oxidation of estradiol (fast rate, moderately inactivating); (2) effect of cumylphenol (intermediate rate, high inactivation) on estradiol (fast rate, moderately inactivating); and (3) effect of phenol (slow rate, non-inactivating) on the oxidation of cumylphenol (intermediate rate, high inactivation). The effect of varying the concentrations of secondary substrates was also investigated, since it was hypothesized that such effects, if any, would be more pronounced when secondary substrates are present in higher concentrations.

In the first scenario, the effect of phenol on estradiol was investigated by comparing the conversion of 10 µM estradiol achieved after 3 h when oxidized alone (control) and in the presence of phenol at initial concentrations ranging from 1 to 50 µM. Note that it has been previously shown that laccase is not measurably inactivated when oxidizing phenol at initial concentrations ranging from 1 to 50 µM [5,6]. A relatively low enzyme concentration of 3.64 nM was used, which was sufficient to ensure a high level of conversion of 10 µM estradiol after 3 h when oxidized alone. The batch reaction results shown in Figure 1a demonstrate that the presence of phenol (P) in molar ratios relative to estradiol (E2) from

1:10 to 5:1 did not significantly affect (Student's *t*-test, $p < 0.05$) the conversion of estradiol compared to the control. From this, several conclusions are made. First, even though it is known that estradiol inactivates the enzyme over the course of the reaction, the presence of phenol as a secondary substrate had no effect (neither positive nor negative) on the rate of inactivation of the enzyme as it oxidized the estradiol. That is, the model predicts that after 3 h of oxidation of 10 µM estradiol by 3.64 nM laccase, approximately 50% inactivation of the initial laccase would occur. Since the conversion after 3 h remained essentially the same in the presence of phenol, it is concluded that this level of inactivation of laccase over the course of the reaction period also did not change. Secondly, it is unlikely that significant quantities of products of a different and inactivating nature were formed during the reaction. For instance, in the experiment with 50 µM phenol shown in Figure 1a, the conversion of the 10-µM estradiol was approximately 90%, whereas only 1.4% of the 50-µM phenol was oxidized (data not shown). Under these conditions, the ratio of phenol radicals to estradiol radicals produced would be approximately 1:13. Thus, the products arising from the oxidation of mixed substrates were somewhat different than those produced during the oxidation of estradiol alone and, furthermore, the limited quantity of mixed oligomeric products that was created were not substantially inactivating or inhibiting in nature. Moreover, it is evident that the oxidized products did not result in an enhancement in the oxidation of the target substrate through free-radical attack. Finally, given that there was insignificant reduction in estradiol conversion in the presence of substantial quantities of phenol, it is apparent that phenol did not competitively occupy the catalytic site of the enzyme while it was being oxidized.

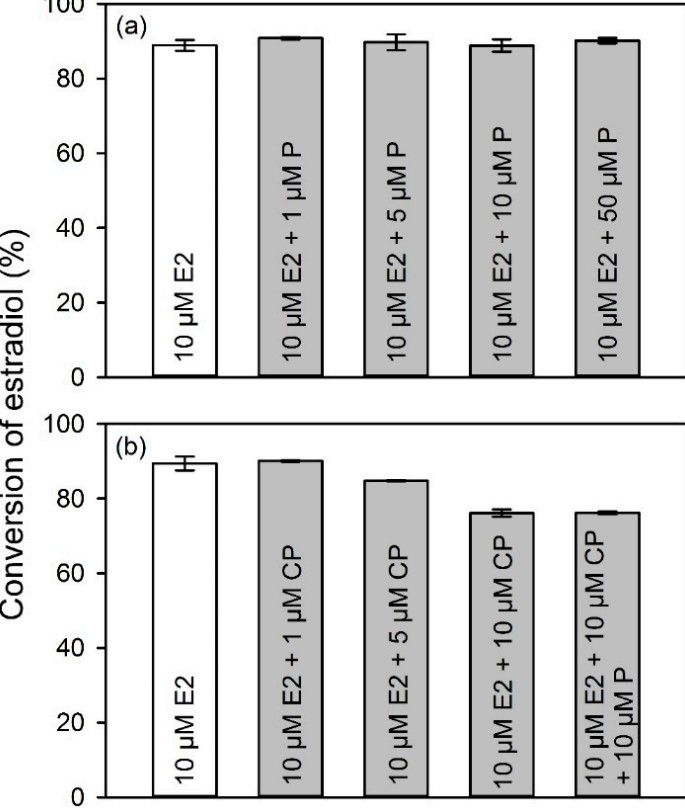

**Figure 1.** Effects of phenol (P; slow rate, non-inactivating) and cumylphenol (CP; intermediate rate, high inactivation) on the conversion of 10 µM estradiol (E2; fast rate, moderate inactivation) after 3 hours (h) in multi-substrate experiments using 3.64 nM laccase: (**a**) effect of 1, 5, 10, and 50 µM phenol (P); (**b**) effect of 1, 5, and 10 µM cumylphenol (CP) and a mixture of 10 µM cumylphenol (CP) and phenol (P).

Additional experiments were conducted to determine whether phenol at a much higher concentration of 500 μM, representing a 50:1 molar ratio of phenol to estradiol, would have a discernable effect. As shown in Figure 2a, the concentration of estradiol was monitored over time in reacting mixtures for a control without phenol and for a reaction conducted in the presence of 50 μM phenol. Due to the very low quantity of enzyme used in these experiments, phenol conversion over time was not significant and is therefore not shown. Consistent with the observations noted above, the conversion of estradiol over time was not impacted by the presence of 50 μM phenol. However, the estradiol conversion was negatively impacted in the presence of 500 μM phenol with only 86% conversion experienced after 3 h in comparison to 95% in the control. As can be seen, the curves overlapped initially and then diverged over time, which is likely due to the gradual inactivation of the enzyme during phenol oxidation, which had been reported at elevated phenol concentrations ranging from 494 to 3410 μM [14], in contrast to the negligible inactivation that occurs at concentrations less than 50 μM [5,6]. Note that this seeming contradiction is not surprising, since the rate of inactivation is both a function of the concentration of the substrate and its rate of oxidation [6,14]. For phenol, which is a very slow substrate with a very low tendency to inactivate laccase, inactivation should only become apparent or even measurable at elevated phenol concentrations.

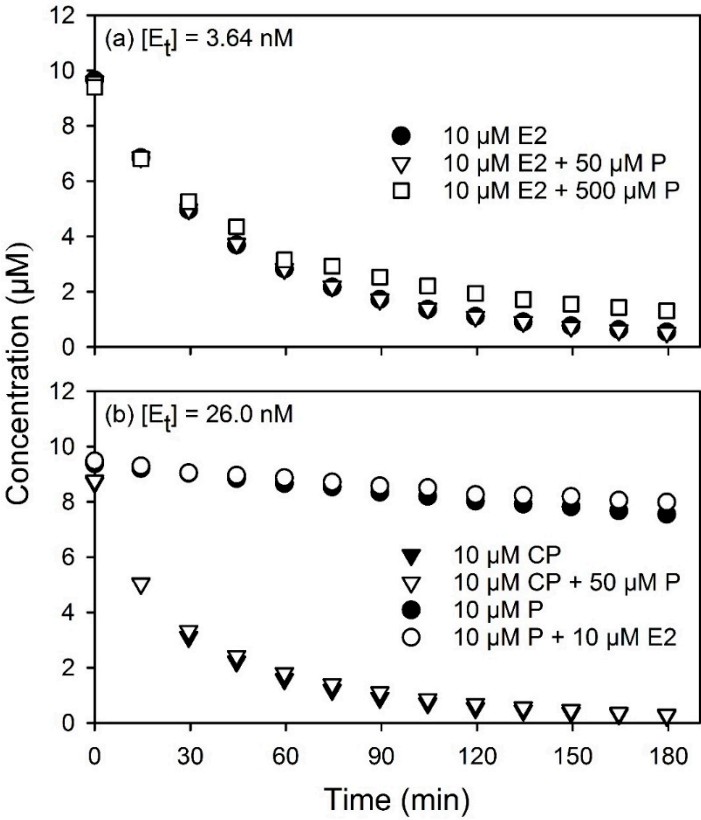

**Figure 2.** Residual concentrations of individual and mixed substrates over 3-h reaction periods: (**a**) effects of 50 and 500 μM phenol (P) on reactions of 10 μM estradiol (E2) with 3.64 nM laccase; (**b**) effect of 50 μM phenol (P) on the reaction of 10 μM cumylphenol (CP) and effect of 10 μM estradiol (E2) on the reaction of 10 μM phenol (P) with 26 nM laccase.

With respect to the second scenario, Figure 1b shows the effect of initial concentrations of 1, 5, and 10 μM cumylphenol (intermediate rate, high inactivation) on the conversion of 10 μM estradiol (fast rate, moderately inactivating). In contrast to the experiments conducted with phenol described above, it was expected that a decrease in estradiol conversion might be observed since: (a) cumylphenol is a much faster substrate than phenol and is kinetically closer to estradiol; and (b) cumylphenol tends to inactivate laccase more than all other substrates studied, and its oxidation over time would

gradually inactivate laccase, thereby negatively influencing the rate of estradiol oxidation. In addition, multi-substrate reactions may produce mixed reaction products, which might have additional impacts on estradiol conversion, especially when a third substrate (i.e., phenol) is present, as will be described below. As shown in Figure 1b, the addition of cumylphenol to estradiol at a ratio of 1:10 had an insignificant effect ($p < 0.05$) on the conversion of estradiol compared to the control. However, for increasingly elevated cumylphenol concentrations at ratios of 1:2 and 1:1 with respect to estradiol, the conversion of estradiol decreased ($p < 0.05$) from 89% in the absence of cumylphenol, to 84% and 76% in the presence of 5 and 10 μM cumylphenol, respectively. These results demonstrate that the slower substrate, cumylphenol, interfered with the conversion of the faster substrate, estradiol, when it was present in comparable concentrations and, furthermore, that the effect increased with increasing cumylphenol concentration. This was likely due to the increase in the rate of inactivation arising from the oxidation of cumylphenol. For example, as mentioned above, when oxidizing 10 μM estradiol alone using 3.64 nM laccase, the single-substrate kinetic model predicts that approximately 50% of the laccase will be inactivated after 3 h. Similarly, if either 5 or 10 μM cumylphenol is oxidized under the same condition, the model predicts that approximately 67% or 78% of the laccase will be inactivated, respectively, after 3 h. Given this, it is not surprising that the cumulative effects of the two sources of inactivation would negatively impact the conversion of 10 μM estradiol in the presence of comparable quantities of cumylphenol. However, these results do not demonstrate conclusively whether the reduced rate of estradiol oxidation is due to: (1) competition for the enzyme by cumylphenol; (2) inactivation arising from the oxidation of cumylphenol; or (3) production of mixed products with inactivating or inhibiting properties. This uncertainty will be addressed below (see Section 2.2).

Also shown in Figure 1b is the impact of the simultaneous presence of both 10 μM cumylphenol and 10 μM phenol on estradiol conversion. In this scenario, the addition of 10 μM phenol did not have any significant effect ($p < 0.05$) on estradiol conversion as compared with that measured for a mixture of 10 μM cumylphenol and estradiol. This agrees with the results shown in Figure 1a, which indicate that mixed reaction products arising from the simultaneous oxidation of phenol and cumylphenol do not have a significant additional impact on estradiol conversion and, therefore, neither contribute to enzyme inactivation/inhibition nor enhanced estradiol conversion through free-radical attack.

In the third scenario, the effect of the presence of a phenol (slow rate, non-inactivating) on the rate of oxidation of cumylphenol (intermediate rate, high inactivation) was tested. Based on the results shown in Figure 1a, it was expected that the effect of phenol on cumylphenol conversion would be minimal unless mixed reaction products were formed with a tendency to inactivate or inhibit the enzyme. Batch reaction results confirmed that there was minimal effect of the presence of phenol on the rate of oxidation of cumylphenol. For example, as shown in Figure 2b, even when 10 μM cumylphenol was oxidized in the presence of 50 μM phenol, there was virtually no difference in the cumylphenol conversion over time in the reaction mixture as compared to the control without phenol. Note that 26 nM laccase was used in these experiments in order to achieve significant cumylphenol conversion in 3 h. From this, it is concluded that the enzyme was not inactivated any more than it was when oxidizing cumylphenol alone and, therefore, no inactivating or inhibiting products were formed due to the mixture of substrates. Furthermore, the enzyme was not significantly distracted from the oxidation of cumylphenol by the presence of the much slower substrate phenol.

Overall, based on the results described above, it is concluded that the presence of one or more secondary substrates that are slow relative to a target substrate have an insignificant or fairly minimal impact on the rate of oxidation of a faster substrate, except when secondary substrate causes enzyme inactivation during its conversion and/or where very substantial quantities of secondary substrates are present. In addition, it appears that the formation of mixed products with inactivating or inhibiting characteristics was not an issue of a concern when oxidizing combinations of estradiol, phenol or cumylphenol.

### 2.1.2. Effects of Fast Secondary Substrates on Slow Target Substrates

Similar to the above, the effects of secondary substrates on the reactions of kinetically slower target substrates were assessed under three scenarios, as will be described below. All experiments were conducted at a fixed initial laccase concentration of 26 nM to ensure that almost-complete conversion of 10 μM cumylphenol and substantial conversion of 10 μM phenol were achieved over 3 h when these substrates were oxidized alone. The results are shown in Figures 2b and 3.

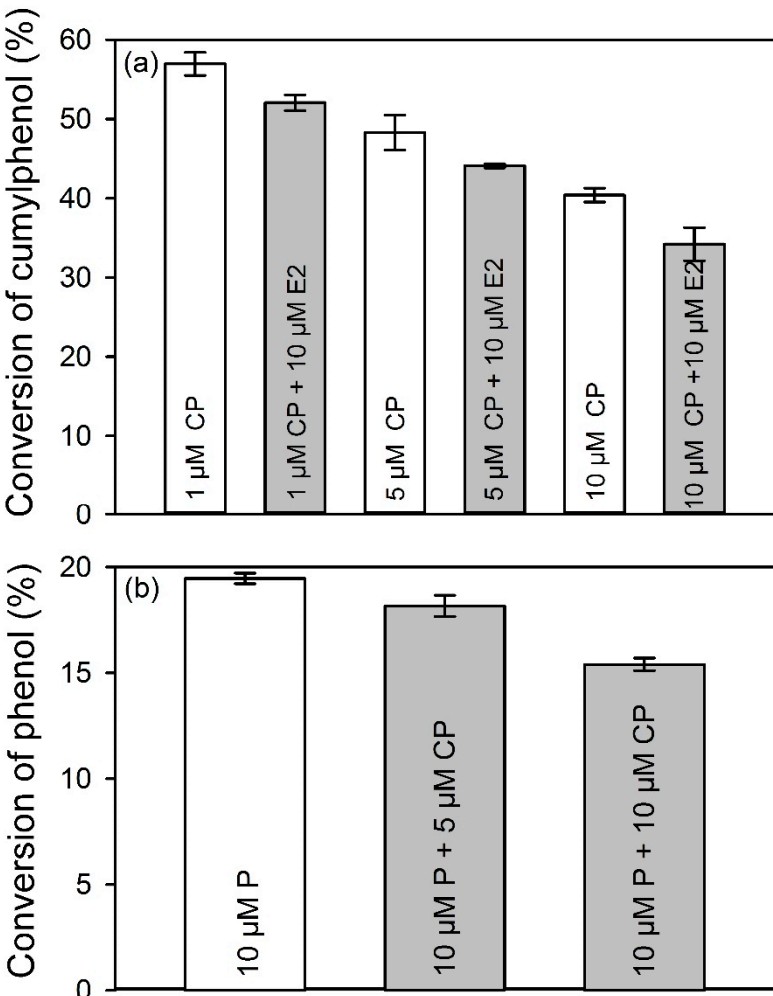

**Figure 3.** Effects of faster substrates on the conversion of slower substrates in multi-substrate experiments using 26 nM laccase after 3 h: (**a**) effect of 10 μM estradiol (fast rate, moderate inactivation) on the conversion of cumylphenol (intermediate rate, high inactivation) with initial concentrations of 1, 5, and 10 μM; (**b**) effects of 5 and 10 μM cumylphenol (intermediate rate, high inactivation) on the conversion of 10 μM phenol (slow rate, non-inactivating).

In the first scenario shown in Figure 3a, the effect of estradiol (fast rate, moderate inactivation) on the target substrate cumylphenol (intermediate rate, high inactivation) was assessed by comparing cumylphenol conversion when oxidized alone at 1, 5, and 10 μM (controls) and in presence of estradiol. A high estradiol initial concentration of 10 μM was selected in order to reveal the maximum impact of this fast substrate, given that estradiol was expected to be rapidly oxidized in the initial minutes of the reaction when using a laccase at a relative high concentration of 26 nM. For example, the single-substrate model predicted that 90% conversion of 10 μM estradiol would be achieved by 26 nM laccase in the first 15 min of the reaction and would result in approximately 19% laccase inactivation. Therefore, inactivation arising from the oxidation of estradiol should negatively impact cumylphenol

conversion as compared with the controls without estradiol. Furthermore, it was anticipated that, in oxidizing the fast substrate first, the oxidation of cumylphenol might be delayed until the estradiol was fully consumed. Finally, it was hypothesized that if any mixed reaction products arising from the simultaneous oxidation of cumylphenol and estradiol had a tendency to inactivate or inhibit laccase, this effect would be more evident at elevated initial cumylphenol concentrations. As shown in Figure 3a, the presence of estradiol had a significant impact ($p < 0.05$) on cumylphenol conversion in all cases, where the absolute difference in conversion between controls and reactions with estradiol was consistently in the range of 4%–6% for experiments where the molar ratio of estradiol to cumylphenol ranged from 10:1 to 1:1. The fact that the decrease in cumylphenol conversion was relatively small over a wide range of molar ratios suggests that the accumulation of mixed reaction products did not dramatically impact enzyme activity. Instead, the decrease of cumylphenol conversion due to the presence of estradiol is more likely to have been due to the additional enzyme inactivation that resulted from estradiol oxidation. However, given that this was not conclusively proven by these experiments, this issue will be addressed below (see Section 2.2).

In the second scenario shown in Figure 3b, the effect of the cumylphenol (intermediate rate, high inactivation) at concentrations of 0, 5, and 10 µM on the reaction of 10 µM phenol (slow rate, non-inactivating) was measured. It was anticipated that, based on the observation that the addition of phenol did not affect the conversion of cumylphenol in experiments shown in Figure 2b, no mixed reaction products capable of inactivating laccase would be formed. However, because the oxidation of cumylphenol causes laccase inactivation, it was anticipated that phenol conversion would be negatively impacted by the presence of cumylphenol, and that this impact would increase with increasing cumylphenol in the reaction mixture. As shown in Figure 3b, the observed average differences in the conversion of phenol in the absence and presence of cumylphenol experiments were minor, albeit statistically significant ($p < 0.05$), with an absolute decrease in conversion of 1.3% and 4.0% when phenol was oxidized in the presence of 5 and 10 µM cumylphenol, respectively. These minor differences suggest that the production of inactivating or inhibiting products was not a factor, and that, as with other experiments described above, the decrease in phenol conversion was most likely caused by laccase inactivation arising from the oxidation of cumylphenol. Note also that since no enhancement in phenol conversion was observed with increasing quantities of cumylphenol in reaction mixtures, this indicates that the free-radical products arising from the oxidation of cumylphenol did not oxidize phenol.

In the third scenario, the impact of 10 µM estradiol (fast rate, moderate inactivation) on the conversion of 10 µM phenol (slow rate, non-inactivating) was assessed. In these experiments, in which 26 nM laccase was used, the residual concentration of phenol was measured over time in the absence and presence of 10 µM estradiol. In this scenario, it was expected that estradiol would be depleted rapidly and would result in partial inactivation of laccase (i.e., when oxidizing 10 µM estradiol alone, the single-substrate model predicts that 99.8% of the 10 µM estradiol initially present would be oxidized in the first 60 min of a reaction, resulting in 26% laccase inactivation). As shown in Figure 2b, the effect of the presence of estradiol in a 1:1 ratio with phenol was miniscule, with only a minor diminution of phenol conversion experienced after 3 h. In this case, it is concluded that inactivation of laccase during the rapid oxidation of the estradiol would account for the slight decrease in phenol conversion. Furthermore, it is concluded that the formation of inactivating or inhibiting products was not significant, and that there was no evidence of enhanced phenol conversion by virtue of the free radicals produced during the oxidation of estradiol.

### 2.1.3. Summary of Results from Mixed Substrate Experiments

Overall, the effect of the presence of secondary substrates on the conversation of a target substrate was a minor decrease in the rate of oxidation. The presence of phenol as a secondary substrate, which is the only substrate used that does not inactivate laccase significantly in the range used for these experiments and which is an extremely slow substrate, did not significantly affect the conversion of either estradiol or cumylphenol (see Figures 1a and 2), except when present in very high quantities (i.e.,

a 50:1 molar ratio of phenol to estradiol). Under the latter condition, the minor but discernible impact observed can be accounted for by the fact that inactivation of laccase does occur when catalyzing the oxidation of high concentrations of phenol [14]. Importantly, these observations suggest that slow secondary substrates do not occupy the active site of the enzyme, thereby distracting or blocking the enzyme from oxidizing the faster target substrates. In turn, this suggests that the time during which a substrate occupies the active site of the enzyme is not the rate-limiting step of the oxidation process and, as a result, substrates in a mixture will tend to be oxidized in parallel, rather than in series.

Also, of the three substrates studied above, the presence of cumylphenol had the largest impact on the rate of oxidation of faster and slower target substrates. This is consistent with the observation that this substrate has the highest tendency to cause laccase inactivation (see Tables A1 and A2). In all experiments, the presence of secondary substrates never increased the conversation of the target substrate, thereby confirming that the free radical products produced during the oxidation of the secondary substrate did not, in turn, oxidize the target substrate. The relatively minor decreases in conversion of the target substrate, even in the presence of substantial secondary substrates, also suggest that the products arising from mixtures of phenol, cumylphenol, and/or estradiol do not have significant inactivating or inhibiting characteristics.

Collectively, these results are very promising in terms of demonstrating the feasibility of targeting the oxidation of substrates in the presence of slow or fast and non-inactivating or inactivating substrates. That is, fast substrates such as estradiol can be targeted for oxidation in the presence of very slow secondary substrates without incurring significant delays or requiring additional enzyme to compensate for the presence of another substrate. Moreover, the presence of rapidly oxidized secondary substrates does not substantially distract the enzyme from the oxidation of slower target substrates. The latter observation is not surprising given that a large amount of enzyme was required to convert the slow target substrate in an acceptable time-frame. This means that when targeting a slow substrate for oxidation there will be an excess of enzyme with respect to the fast secondary substrate, which is oxidized at an extremely fast rate relative to the slow substrate. Given these observations, the evidence indicates that secondary substrates that cause enzyme inactivation will have a more important negative impact on the conversion of the target substrate whether they are slow or fast substrates. However, in such instances, the reduced rate of oxidation of the target substrate can be compensated for either by using more enzyme or by lengthening the reaction time to achieve the required level of conversion, as will be shown in Section 2.2.3, below.

## 2.2. Modeling the Kinetics of Mixed Substrates

The single-substrate model developed earlier [5,6] to describe the oxidation kinetics of individual substrates was adapted here in order to extend its application to substrate mixtures. This was done in order to provide further insight into the observations made above, and to serve as a tool for evaluating the feasibility of using laccase to target a particular substrate for oxidation in the presence of other secondary substrates.

Based on the observations made above of mixed substrate experiments of estradiol, cumylphenol, and phenol, three assumptions were made for the purposes of kinetic modeling. First, it was assumed that all substrates in mixtures will react with laccase at rates described by the kinetic parameters that were evaluated when they were oxidized alone. This is equivalent to assuming that the presence of secondary substrates neither enhances nor detracts from the oxidation of the target substrate through synergistic (e.g., free-radical attack) or antagonistic (e.g., inhibiting) effects. Secondly, it was assumed that inactivation arising from the oxidation of individual substrates occurs independently of the oxidation of other substrates and is therefore additive (i.e., neither synergistic nor antagonistic). Thirdly, it was assumed that mixed products formed through the combination of different substrate radicals (i.e., mixed substrate polymers and/or radicals produced through their laccase-catalyzed oxidation) neither inactivate nor inhibit laccase.

In light of these assumptions, the development of the multi-substrate model is presented in Appendix A, where it is shown that for a mixture of *N* substrates, the model is comprised of *N* + 4 equations, as listed below, where Equation (1) is an enzyme mass balance equation, Equation (2) describes the rate of oxygen consumption through the catalytic reaction and its replenishment through mass transfer, Equation (3) describes the rate of production and consumption of native enzyme in the catalytic cycle, Equation (4) describes the total rate of enzyme inactivation arising from the oxidation of each of the *N* substrates, and Equation (5) must be written *N* times to describe the rate of oxidation of each of the *N* substrates.

$$[E^*] = [E_t] - [E] - [E_i] \tag{1}$$

$$\frac{d[O_2]}{dt} = -k_1[E][O_2] + k_L a([O_2]_{sat} - [O_2]) \tag{2}$$

$$\frac{d[E]}{dt} = -k_1[E][O_2] + \sum_{n=1}^{N} k_{s_n}[E^*]^{\alpha_n}[S_n]^{\beta_n} \tag{3}$$

$$\frac{d[E_i]}{dt} = ([E_t] - [E_i]) \sum_{n=1}^{N} k_{i_n} \left(-\frac{d[S_n]}{dt}\right)^{0.5} \tag{4}$$

$$\frac{d[S_n]}{dt} = -k_{s_n}[E^*]^{\alpha_n}[S_n]^{\beta_n} \tag{5}$$

In the above equations, at any time, *t*, after the initiation of the reaction, *[O₂]* is the concentration of oxygen, *[Sₙ]* is the concentration of substrate *n* in a mixture of *N* substrates, *[E]* is the concentration of laccase in the native reduced state, *[E\*]* is the concentration of laccase in the oxidized state, *[Eᵢ]* is the concentration of laccase in an inactivated state, and *[Eₜ]* is the total concentration of laccase added to the reaction mixture. The parameter $k_1$ is a kinetic constant describing the rate of oxidation of laccase by oxygen, and $k_L a$ is a mass transfer coefficient for oxygen diffusing from ambient air into a batch-reaction mixture with an oxygen saturation concentration of *[O₂]ₛₐₜ*. In addition, each substrate, *n*, has its own kinetic parameters including a rate constant, $k_{s_n}$, and two exponents, $\alpha_n$ and $\beta_n$, as well as a rate constant, $k_{i_n}$, describing its tendency to inactivate laccase.

These parameters are described in detail in Appendix A, and the theoretical and experimental bases for the development of the model equations are described elsewhere [5,6]. Note that the multi-substrate model was not calibrated against experimental data collected in the present study, since all kinetic parameters for the individual substrates and the general model parameters had been reported earlier [6] for these substrates under identical reaction conditions, as summarized in Table A1.

The predictive ability of the multi-substrate model was tested by comparing model simulations to data collected from experiments where the concentrations of substrates in reaction mixtures were measured over time. In this manner, the success or failure of the multi-substrate model to describe the simultaneous conversion of various substrates could be used to test the validity of the assumptions used in its development. For purposes of comparison, single-substrate experiments were performed in parallel with mixed-substrate experiments, and the single-substrate kinetic model (which is equivalent to the multi-substrate model applied to a single substrate) was used to model the time course of these reactions. Experiments were initially focused on the mixed reactions of phenol, cumylphenol, and estradiol. A fourth substrate, triclosan, was introduced in subsequent experiments to further test the model's predictive abilities and to test the general validity of the assumptions used in its development.

### 2.2.1. Reactions of Phenol, Cumylphenol, and Estradiol

Single- and multi-substrate experiments were performed at laccase concentrations of 3.64 and 26 nM for phenol, cumylphenol, and estradiol at initial substrate concentrations in the vicinity of 10 µM. Measurements of substrate concentration over time are shown for single-substrate reactions in Figure 4a,c, and for multi-substrate reactions in Figure 4b,d. The corresponding model predictions

are indicated by the solid lines, where the model equations were solved numerically using the initial conditions of substrate and laccase concentrations corresponding to each experiment.

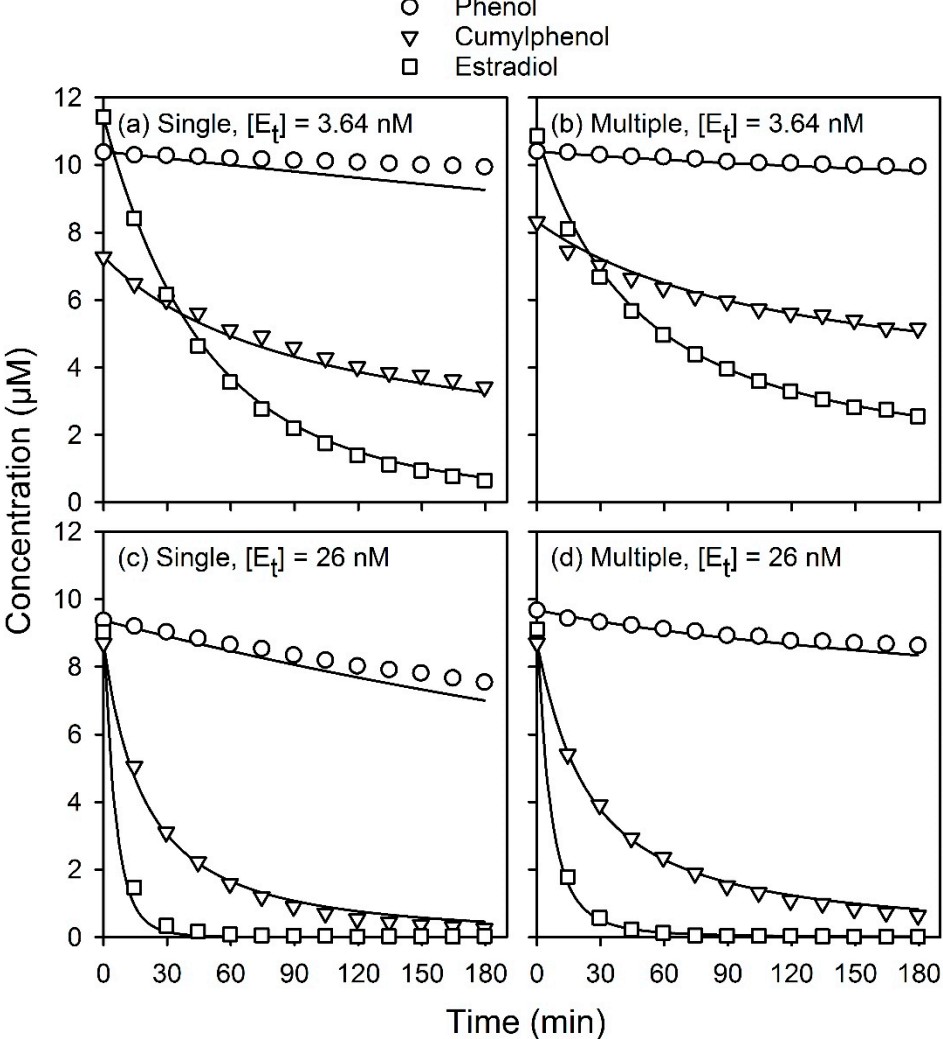

**Figure 4.** Experimental measurements (symbols) and model predictions (lines) for the laccase-catalyzed oxidation of phenol, cumylphenol, and estradiol over a 3-h reaction period: (**a**) oxidation of individual substrates using 3.64 nM laccase; (**b**) oxidation of a mixture of all three substrates using 3.64 nM laccase; (**c**) oxidation of single substrates using 26 nM laccase; (**d**) oxidation of a mixture of all three substrates using 26 nM laccase.

The excellent correspondence between measured and predicted substrate concentrations over time in Figure 4 indicates that the single- and multi-substrate models accurately predicted the time course of reactions of phenol, cumylphenol, and estradiol at low (3.64 nM) and intermediate (26 nM) laccase concentrations over a 3-h reaction period. When compared with the predictions of the single-substrate model, the multi-substrate model performed equally well, or even better, in predicting the time course of reactions of these three mixed substrates. Consistent with the results shown in Figures 1–3, a comparison of Figure 4a,b shows that the conversions achieved over time of substrates in mixtures were less than when substrates were oxidized alone, especially for estradiol and cumylphenol. The success of the model in predicting these differences shows that it is indeed the cumulative inactivation arising from the oxidation of individual substrates that primarily causes the slowing in the conversion of the substrates when they are present in mixtures.

A similar slowing of multi-substrate reactions is also evident when regarding experiments with laccase concentrations of 26 nM in Figure 4c,d. In this case, the slowing of the multi-substrate reactions compared to single-substrate reactions appears to be less pronounced than for experiments conducted at 3.64 nM laccase, except for phenol. This is not unexpected, since at a higher laccase concentration the faster substrates are rapidly depleted before substantial inactivation occurs, but as phenol continues to be slowly oxidized, the impact of cumulative inactivation becomes more apparent over time. The most significant, albeit minor, deviations between model prediction and experimental results were observed for phenol, especially as seen in Figure 4a,c. This is consistent with observations reported in an earlier work [6], where model predictions for phenol had a tendency to slightly over-predict reaction rates for situations with high phenol and relatively low laccase concentrations. Note that kinetic parameter calibrations for phenol were performed at much higher laccase concentrations (i.e., 120–2500 nM) [5,6] than those used in the present study, and, as such, minor deviations were not unexpected.

The success of the multi-substrate model in accurately predicting the time course of mixed substrate reactions without any additional calibration confirms that the cumulative and substantial inactivation of laccase during the simultaneous oxidation of compounds caused the reduced rate of oxidation of substrates in mixtures. This also confirms that no enhancements in substrate conversion were caused through free-radical attack reactions and, furthermore, the mixed reaction products did not cause inhibition or inactivation of laccase. Therefore, for the substrates phenol, cumylphenol, and estradiol, the validity of the assumptions made during model development are confirmed. To our knowledge, this is the first report of a kinetic model that has the capability to accurately model the simultaneous enzymatic oxidation of mixtures of substrates over prolonged reaction times.

2.2.2. Reactions of Triclosan, Phenol, Cumylphenol, and Estradiol

The successful modeling of reactions of mixed substrates described above motivated the broadening of the application of the multi-substrate model to include a fourth substrate, triclosan, whose kinetic parameters had been estimated in previous work under identical reaction conditions [6] (see Table A1). Triclosan is a substrate with an intermediate rate of oxidation and is moderately inactivating (see Table A2). Batch reactions of four substrates were performed in the same manner as those described above and using 3.64 nM laccase.

Experiments were first conducted with initial substrate concentrations in the vicinity of 10 μM. Figure 5a,b show the experimental observations and corresponding model predictions for reactions of single and mixed substrates, respectively. As shown in Figure 5a, the single-substrate model accurately predicted the time course of the reactions of single substrates. However, as shown in Figure 5b, the multi-substrate model failed when predicting the reaction kinetics for a mixture of triclosan, cumylphenol, estradiol, and phenol. Note that, in this case, the predictions for phenol appear to be accurate, but this is because phenol conversion using a low laccase concentration of 3.64 nM is very slow and, as a result, the differences between experimental measurements of concentration over time and model results are expected to be small.

When comparing the experimental results in Figure 5a,b, it can be seen that the conversion of individual substrates observed in mixtures are substantially reduced as compared with that achieved when they were oxidized alone. This is consistent with the second assumption made during model development, where it was expected that the addition of triclosan, which is a moderately inactivating substrate (see Table A2), would slow the conversion of the other substrates in the mixture. However, the multi-substrate model, which accounts for this additional inactivation, consistently over-predicted the conversion of triclosan, cumylphenol, and estradiol over time. This would imply that the introduction of triclosan into the reaction mixture caused an additional source of inactivation that is not accounted for in the model. This observation invalidates the third assumption used in model development for this multi-substrate reaction involving triclosan.

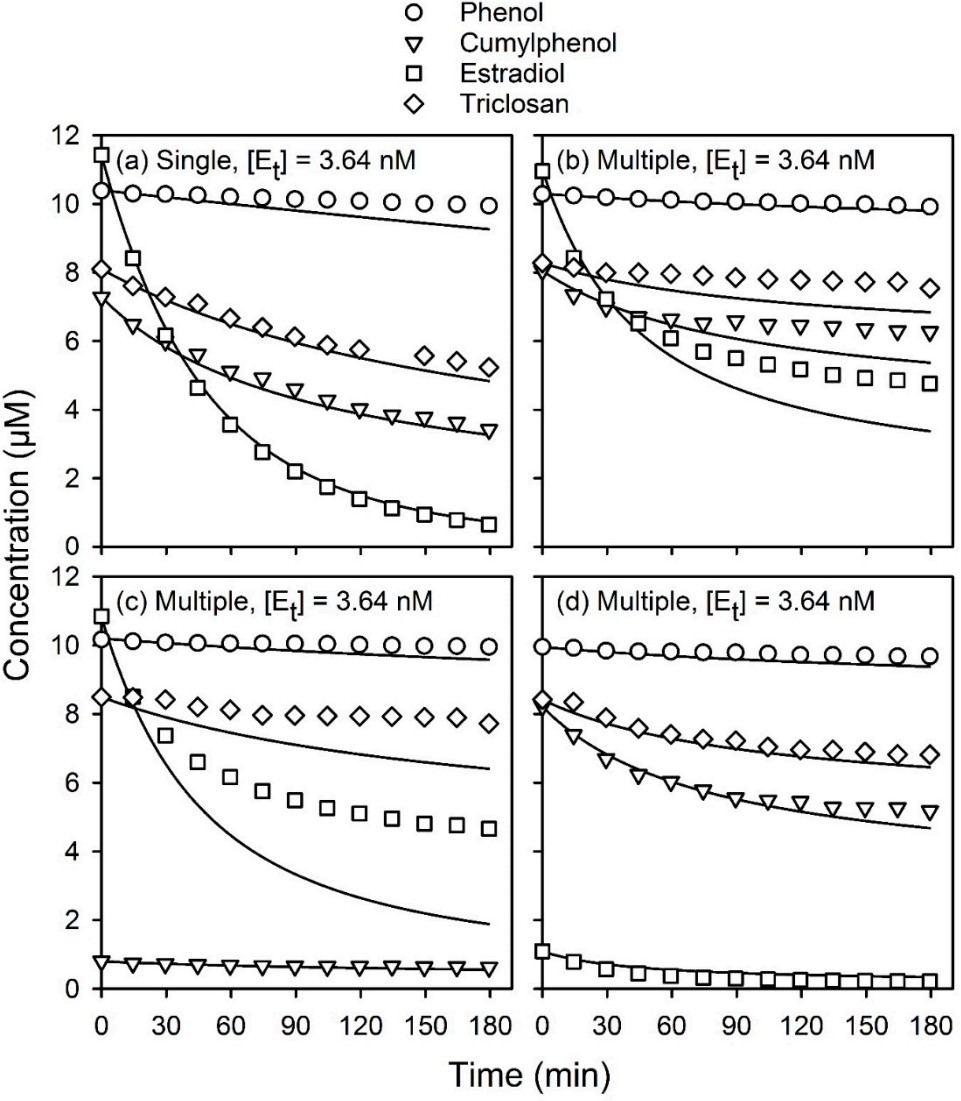

**Figure 5.** Experimental measurements (symbols) and model predictions (lines) for the laccase-catalyzed oxidation of phenol, cumylphenol, estradiol, and triclosan over a 3-h reaction period using 3.64 nM laccase: (**a**) oxidation of individual substrates alone; (**b**) oxidation of a mixture of all four substrates with elevated cumylphenol (8 μM) and estradiol concentrations (11 μM); (**c**) oxidation of a mixture of all four substrates with a low initial cumylphenol concentration (1 μM); (**d**) oxidation of a mixture of all four substrates with a low initial estradiol concentration (1 μM).

In order to identify the source of this additional inhibition or inactivation, experiments were conducted where the initial concentrations of selected substrates were varied. In conducting these experiments, it was assumed that if mixed reaction products are formed between triclosan and another substrate that causes inactivation, then changes in the concentrations of either of these substrates should cause varying levels of divergence between model predictions and experimental results. Given that phenol conversion is minor at the low laccase concentration (3.64 nM) used in these experiments, focus was placed on investigating the possible formation of mixed reaction products arising from the simultaneous oxidation of triclosan and estradiol and/or triclosan and cumylphenol.

In the first experiment, the initial concentration of cumylphenol was reduced to approximately 1 μM, while other initial substrate concentrations were maintained at the levels shown in Figure 5b. As can be seen in Figure 5c, under this condition of low initial cumylphenol concentration, the conversion of triclosan and estradiol were substantially over-predicted by the multi-substrate model with errors greater than those observed in Figure 5b. The substantial increase in conversion that was predicted for

triclosan and estradiol over time in Figure 5c, in comparison with those predicted in Figure 5b, was expected to occur, since the highly inactivating cumylphenol is present in a much lower concentration and, therefore, less inactivation was predicted. However, as can be seen by comparing the experimental measurements in Figure 5b,c, the reduced cumylphenol concentration did not result in an anticipated substantial increase in conversion of estradiol or triclosan. This would imply that the products created during the simultaneous oxidation of triclosan and estradiol are a significant source of laccase inactivation or inhibition.

In a second experiment, the initial concentration of estradiol was reduced to approximately 1 μM, while the initial concentrations of other substrates were the same as those used in the experiment pictured in Figure 5b. As shown in Figure 5d, the predictions of the multi-substrate model for this reaction were very accurate for all substrates, with only minor errors observed between experimental measurements and model predictions after the second hour for cumylphenol and triclosan. This suggests that the mixed reaction products of cumylphenol and triclosan did not significantly contribute to additional enzyme inactivation or inhibition. In this case, the growing error between modeled and experimental results after 2 h are most likely due to the formation of mixed estradiol–triclosan reaction products in much lower quantities than those experienced in the experiment of Figure 5b, because the initial molar ratio of estradiol and triclosan was only 1:7.6, rather than 1:0.75.

It is concluded from the above that the multi-substrate model failed to accurately predict substrate concentrations over time for a mixture of phenol, triclosan, cumylphenol, and estradiol, due to products arising from the simultaneous oxidation of triclosan and estradiol. This suggests that the mixed-substrate model can be applied with confidence to reactions of mixtures of phenol, cumylphenol, and triclosan when estradiol is not present. This was verified by conducting further experiments involving these three substrates with laccase concentrations of 3.64 and 26 nM. Figure 6a,c show measured and modeled results for reactions of these substrates when oxidized alone and Figure 6b,d show corresponding results for reactions of mixed substrates under the same conditions. As can be seen, the single-substrate and mixed-substrate models perform equally well in predicting the time course of the reactions of phenol, triclosan, and cumylphenol.

Collectively, these results indicate that the simultaneous oxidation of triclosan and estradiol in mixtures results in inactivation or inhibition of laccase at levels that are not accounted for by the model. Therefore, when triclosan and estradiol are present, the third assumption used in the development of the mixed substrate model is not valid. Consequently, in order to accurately model reactions involving the simultaneous oxidation of triclosan and estradiol, it will be necessary to incorporate an expression into the model than can account for this phenomenon. Furthermore, in any given reaction system involving multiple substrates, and not just those used here, it will be necessary to conduct experiments such as those shown in Figure 5 to determine if such phenomena represent an additional source of inactivation or inhibition that might limit the application of the multi-substrate model or, more importantly, undermine the use of laccase to target the oxidation of one or more substrates in a mixture.

### 2.2.3. Application of the Multi-Substrate Model

As shown in Figures 4 and 6, the multi-substrate model can be used to predict the transient kinetics of reactions of mixtures of aqueous phenolic substrates, except in the special case where certain combinations of substrates (e.g., estradiol and triclosan) result in an additional source of inactivation or inhibition of laccase. The model can also be used to gain insights into phenomena involved in reactions of multiple substrates, and to assess the feasibility of applications where the goal is to oxidize one or more targeted substrates in a mixture of other substrates.

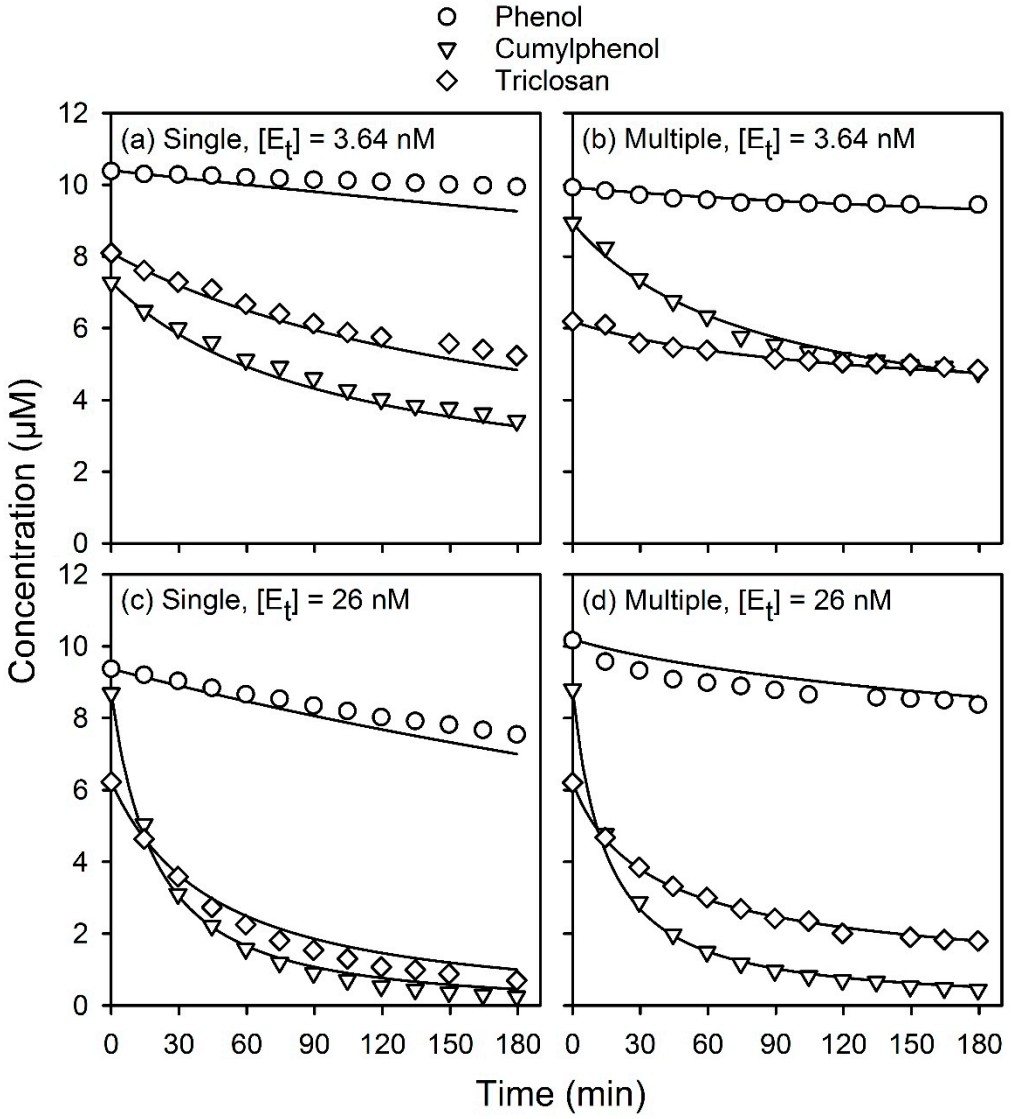

**Figure 6.** Experimental measurements (symbols) and model predictions (lines) for the laccase-catalyzed oxidation of phenol, cumylphenol, and triclosan: (**a**) oxidation of individual substrates using 3.64 nM laccase; (**b**) oxidation of a mixture of all three substrates using 3.64 nM laccase; (**c**) oxidation of individual substrates using 26 nM laccase; (**d**) oxidation of a mixture of all three substrates using 26 nM laccase.

For instance, one can use the model to independently investigate the effects of secondary substrates with differing rates of oxidation and tendencies to cause inactivation—something that may be difficult to do experimentally. In such simulations, the effect of the magnitudes of the oxidation rate constants, the inactivation rate constants, and the concentrations of secondary substrates can be investigated independently. Furthermore, the model can be used to approximate the distribution of enzyme amongst various states (i.e., E, E*, $E_i$) over time under different reaction conditions, which can be used to gain insights into competing phenomena that influence the reaction rate of the target substrate. Examples of such simulations are demonstrated in Appendix C, using the multi-substrate model where the impacts of the magnitude of the initial concentration and inactivation rate constants of hypothetical secondary substrates that are kinetically similar to the target substrate are investigated for estradiol and cumylphenol. When observations from such simulations are combined with experimental observations discussed above, the following general statements can be made about the oxidation of target substrates in the micromolar or sub-micromolar concentrations range in the presence of a secondary substrate: (1) non-inactivating secondary substrates will only have a significant negative impact on the rate

of oxidation of the target substrate if they are both very fast substrates of laccase and, furthermore, this impact will increase with an increasing concentration of the secondary substrate relative to the target substrate; and (2) inactivating secondary substrates of a comparable or faster rate than the target substrate will have an important negative impact on the oxidation of the target substrate, and these impacts will increase with the tendency of the secondary substrate to cause inactivation, as well as with its concentration.

Beyond the insights gained above, the model can also be used to assess the feasibility of targeting substrates individually and in mixtures for oxidation. Typically, feasibility will be defined in terms of the quantity of enzyme and reaction time required to achieve a given level of conversion in a selected reactor configuration, all of which have important implications with respect to the operating and capital costs of the reactor systems. In light of this, the utility of the multi-substrate model is demonstrated below by showing how it can be applied to assess the feasibility of targeting the oxidation of estradiol, cumylphenol, and triclosan in mixtures of other substrates in batch reactors. The oxidation of phenol as the target substrate in mixtures is not investigated here because it was concluded previously that this compound cannot be targeted for oxidation at micromolar or sub-micromolar concentrations without using excessive quantities of enzyme or overly long reaction times [5,6]. However, its impact as a secondary substrate will be investigated below.

The first demonstration of the model's utility is shown in Figure 7a,c,e. In these figures, the model has been used to estimate the reaction time required to achieve 90% and 99.9% conversion of a range of initial concentrations of triclosan (TCL), cumylphenol (CP), and estradiol (E2), respectively, in the presence of 10 µM secondary substrates while using an initial laccase concentration of 125 nM. This quantity of enzyme was selected because it was used in previous work when investigating the oxidation of single substrates [6] and represents an intermediate concentration of enzyme equal to approximately 8.2 mg·L$^{-1}$ based on a laccase molecular mass of 65 kDa. These concentrations are within the range of those used to calibrate the model's kinetic parameters (see Table A1). For each of the targeted substrates, 90% and 99.9% conversion lines are plotted for the target substrate alone (e.g., TCL in Figure 7a), for the target substrate in the presence of a secondary substrate (e.g., TCL + 10 µM P; TCL + 10 µM CP), and for the target substrate in the presence of two secondary substrates (e.g., TCL + 10 µM CP + 10 µM P). Note that the dotted portions of the lines indicate when simulations were conducted outside of the range of substrate concentrations used to calibrate the model [6], where they may be subject to greater error. Due to the findings reported in Section 2.2.2, the model was not used to simulate reactions when estradiol and triclosan were present simultaneously.

As can be seen in Figure 7a,c,e, the curves corresponding to reactions of substrate alone and substrate in the presence of 10 µM phenol completely overlap. This indicates that phenol, which is a slow and non-inactivation secondary substrate, has insignificant impact at this concentration on the reaction times required to achieve various levels of conversion of the substrates. As indicated by the shift of the curves to the right when reactions are conducted in the presence of other substrates, the secondary substrates consistently slow the reaction of the target substrate, both by distracting the enzyme from the target substrate, and by inactivating the enzyme during the oxidation of the secondary substrate. The effect of enzyme inactivation is evident in these figures through the C-shaped curvature of the conversion lines, especially for the lines corresponding to 99.9% conversion. This curvature results from competing phenomena in a reaction mixture that vary in importance as a function of substrate concentration. That is, at high initial concentrations of the target substrate, a high rate of oxidation is favored. However, under this condition, the rate of inactivation arising from the oxidation of the target compound is also favored. Overall, these competing phenomena result in a slower average rate of reaction, and an increase in the reaction time required to achieve a desired level of conversion. However, as the initial concentration of the target substrate is reduced, the rate of inactivation decreases, resulting in a reduced reaction time. Then, when the initial concentration is reduced further, the rate of reaction of the target substrate becomes mainly limited by the availability of the substrate, thereby lengthening the reaction time required. According to all of these figures, 90%

conversion of a wide range of initial substrate concentrations can be achieved using 125 nM laccase with reaction times on the order of hours, even in the presence of other secondary substrates. Moreover, 99.9% conversion often appears achievable with reasonable reaction times, especially in the case of estradiol, which has a very fast reaction rate.

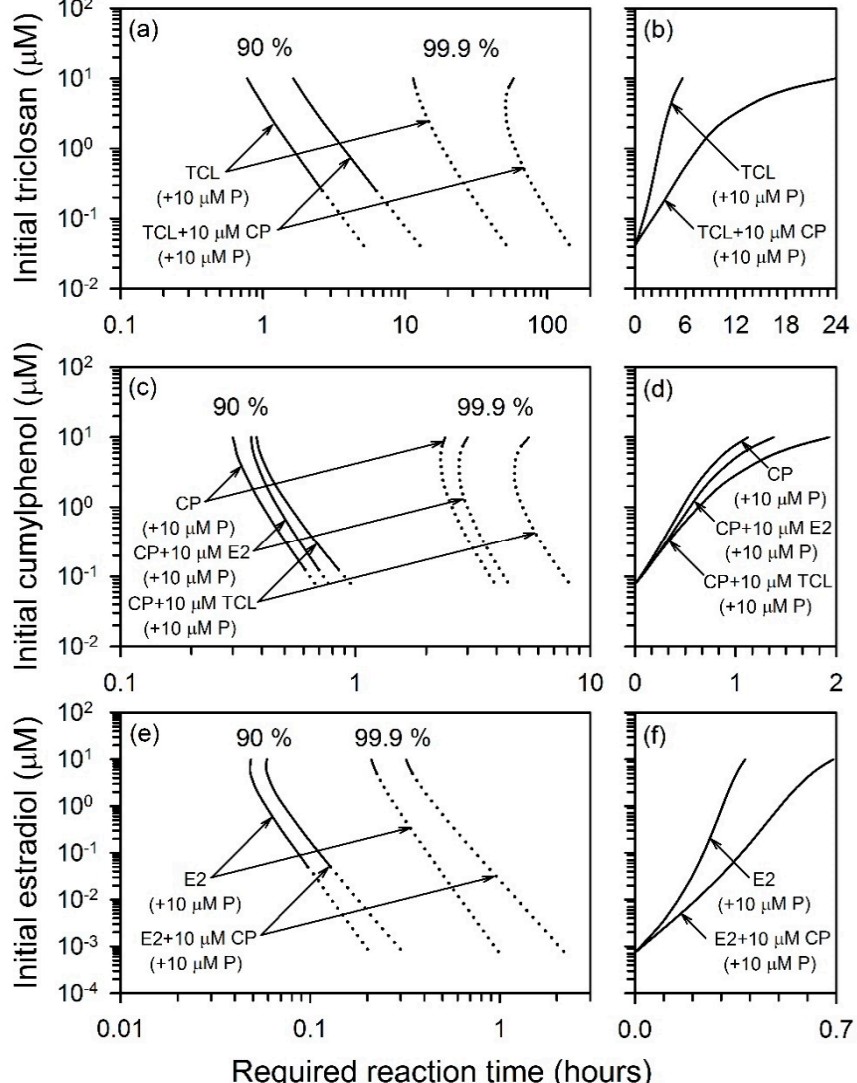

**Figure 7.** Model predictions of (1) reaction times required to achieve 90% or 99.9% conversion of target substrates in batch reactions for a range of initial target substrate concentrations, both with and without the presence of 10 μM secondary substrates (**a,c,e**); and (2) reaction times required to achieve final concentrations of the targeted substrate equal to 100 times the predicted no-effect concentration (PNEC) (**b,d,f**). Model predictions are shown for reactions of triclosan (TCL), cumylphenol (CP), and estradiol (E2) both as target and secondary substrates in the presence of 125 nM laccase. The insignificant effect of phenol (P) as a secondary substrate is also shown. Solid portions of lines represent predictions within the range of calibration/verification of the model, and dotted lines represent predictions outside of this range.

A second demonstration of the model's utility is illustrated in Figure 7b,d,f where the model was used to estimate the time required to achieve a selected final concentration of the targeted substrate in a reaction mixture. As reported previously [6], the final concentrations in a wastewater effluent were selected based on each substrate's respective predicted no-effect concentration (PNEC), which is designed to protect aquatic organisms, and by applying a factor of 100 to account for a moderate

level of dilution of wastewaters after they are deposited into a receiving surface water body [15]; thus, the desired final concentration in treated effluent will be 100 times the PNEC. The PNEC values for estradiol and triclosan have been reported as 7.3 pM [16] and 0.40 nM [17], respectively, and that of cumylphenol is estimated to be 0.77 nM [6]. Therefore, the desired final concentrations of $100 \times$ PNEC for triclosan (40 nM) and cumylphenol (77 nM) were within the range of calibration of the model for these substrates. However, that of estradiol (0.73 nM) was almost one order of magnitude lower than the lowest range of validation of the model [6]. However, given that it has been shown that the model can accurately make predictions far outside of its range of calibration [5,6], model predictions were performed here with a high degree of confidence. Once these values were selected, the multi-substrate model was used to estimate the reaction times required to achieve desired final concentrations for a range of the initial substrate concentrations using 125 nM laccase, as well as in the absence and presence of 10 µM secondary substrates.

The effect of inactivation is seen in the S-shape of the curves in Figure 7b,d,f, where a faster rate of inactivation of the target or secondary substrate results in the elongation of the curves toward longer reaction times for high initial substrate concentrations. The impact of the presence of secondary substrates is revealed through a shift of curves to the right toward longer reaction times, except for phenol which had no impact on any of the reactions simulated. The magnitude of the shift is dependent on the rate of oxidation of the secondary substrate and its tendency to inactivate laccase. For example, Figure 7b shows that the oxidation of triclosan to the desired level of conversion in the presence of 10 µM cumylphenol takes much longer than when oxidized alone, but, for initial concentrations of triclosan below 1 µM, the desired final concentration can be achieved in less than 6 h with 125 nM laccase. Also, it can be seen in Figure 7d that the desired level of conversion of cumylphenol, starting with an initial concentration of 7.5 µM, can be achieved in approximately 1 h in batch reactions with 125 nM laccase when oxidized alone, and in approximately 1.6 h in the presence of 10 µM triclosan, whereas the presence of 10 µM estradiol only increases the required reaction time to 1.2 h. In contrast, as shown in Figure 7f, the oxidation of estradiol to the desired concentration is achieved with much shorter reaction times, with some lengthening of the required reaction times due to the presence of cumylphenol, but not phenol. Note that if the application of the model is extended further, far outside of the range of its calibration and what is shown on Figure 7f, it is predicted that the time required to achieve a final concentration of estradiol of 73 pM (i.e., 10 times its PNEC) for 10 µM estradiol is 0.6 h, but that it increases to approximately 1.3 h in the presence of 10 µM cumylphenol.

The model can also be used to construct similar graphs for other laccase concentrations, for other concentrations of targeted and secondary substrates, and for other reactor configurations (e.g., continuous stirred tank reactors, plug flow reactors). The implementation of such a model in other reactor configurations was demonstrated previously for a model of the oxidation of phenol using horseradish peroxidase [18]. Such simulations could be used to determine the optimum combination of reactor configuration, laccase concentration, and reaction time that would result in the lowest capital and operating costs. Furthermore, when the presence of a secondary substrate negatively impacts the rate of conversion of a targeted substrate, the model can be used to calculate how much additional enzyme or lengthened reaction time would be required to overcome these effects in order to achieve a desired final concentration.

## 3. Materials and Methods

The materials and methods used for all experiments, analyses, and kinetic modeling are summarized below. Many details have been reported previously [5,6] and will not be repeated here.

### 3.1. Materials

Laccase (ES 1.10.3.2) from *T. versicolor* (with a nominal activity of 23.1 U/mg, as per the supplier), 17β-estradiol (98%), 4-cumylphenol (99%), and [2,2′-azino-bis-(3-ethyl-benzothiazoline-6-sulfonic acid)] diammonium salt (ABTS, 98%) were obtained from Sigma-Aldrich (St. Louis, MO, USA).

Triclosan (99%) was obtained from Alfa Aesar (Hard Hill, MA, USA). Sodium phosphate dibasic anhydrous (American Chemical Society (ACS) grade) and acetonitrile (High Performance Liquid Chromatography (HPLC) grade) were obtained from Fisher Scientific Canada. Sulfuric acid (ACS grade) and citric acid anhydrous (ACS grade) were obtained from Acros (NJ, USA). Ultrapure water was prepared using a reverse-osmosis installation from Biolab Equipment Ltd. (Dorval, QC, Canada).

All reagents, batch reaction mixtures, and assay solutions were prepared using citrate-phosphate buffer (CPB) at pH 5.0 with an ionic strength of 0.10. Laccase stock solutions were prepared with 10 mg·mL$^{-1}$ of dry laccase in CPB and filtered through 0.2 μm IC polytetrafluoroethylene (PTFE) Millex membrane filters with low protein binding from Millipore (Billerica, MA, USA). All stock solutions of phenolic substrates (1.0 mg·mL$^{-1}$) and their corresponding working standard solutions were prepared in methanol. Standards were prepared by adding 10 μL of working solutions to up to 1.0 mL of CPB (with a methanol content of 1%) containing deactivated laccase at the concentrations used in batch reactions and previously acidified to pH 2. ABTS solutions (2.0 mM) were prepared using ultrapure water. All stock solutions, working solutions and CPB were stored at 4 °C and equilibrated to 25 °C before use.

### 3.2. Reactant Concentration Measurements

The activities of enzyme stock solutions and reaction mixtures were determined using a colorimetric assay based on the laccase-catalyzed oxidation of ABTS, where the average rate of increase of light absorption at 420 nm was measured at 25 ± 0.05 °C over a period of 1 minute at 0.5 second intervals [5]. The 1.00-mL assay mixture in the cuvette contained approximately 0.5 μg·mL$^{-1}$ laccase and 80 μM ABTS. The assay reaction was initiated through the addition of an aliquot of ABTS stock solution. Laccase activity, *A*, was expressed in units per mL (U·mL$^{-1}$), where 1 U is defined as the amount of enzyme that converts 1 μM of ABTS per minute using an extinction coefficient, $\varepsilon$, of 36 mM·cm$^{-1}$ [19]. The molar concentration of laccase [Et] (μM) was then estimated using a proportionality factor, $f$ = 0.130 ± 0.001 μM·U$^{-1}$·mL, reported previously [5], which directly relates laccase concentration to enzyme activity *A* (U·mL$^{-1}$).

Substrate concentrations in samples were determined using a Model 1100 high performance liquid chromatograph (HPLC) from Agilent Technologies (Santa Clara, CA, USA), equipped with a fluorescent, diode array and multiple wavelengths detectors on XB-C18 Kinetex column (2.6 μm, 100 A, 50 × 3.00 mm), and a 0.5-μm inline filter (Phenomenex, Torrance, CA, USA). A reverse-phase chromatography gradient method (with the organic phase ranging from 25% to 90%) was used with water and acetonitrile as the mobile phases, a flow rate of 0.4 to 1.0 mL·min$^{-1}$, and an injection volume of 20 μL. The auto-sampler tray temperature was maintained at 4 °C and the column temperature at 35 °C. Concentrations were monitored using excitation/emission wavelengths of 270/308 nm for phenol, 226/312 for estradiol, and 227/312 nm for cumylphenol. The concentration of triclosan was monitored using a high-pressure micro cell (1.7 μL; 6 mm) at 200 nm at bandwidth 4 using a reference of 360 nm, response time 0.5 s. and slit 4 nm. The retention times for phenol, estradiol, cumylphenol, and triclosan were 0.8, 1.4, 2.5, and 2.9 min, respectively.

All analytical methods for phenolic substrates, including sample preparation and instrumental analysis, were validated against Food and Drug Administration (FDA) standard methods [20], including: lower limit of quantification (LLOQ) of 5 nM for phenol, estradiol, and triclosan, and 20 nM for cumylphenol; linearity in the LLOQ range to 12 μM; quality controls; dilution integrity; matrix effect; mixed products interference; stock and sample stability at room temperature; injection medium stability and stock stability at 4 °C; and inter-experimental precision and accuracy.

### 3.3. Batch Reaction Experiments

The initial concentrations of estradiol, cumylphenol, triclosan, and phenol in batch reactions ranged from 1 to 12 μM, either alone or when in mixtures. For all reactions, an initial laccase concentration of either 3.64 or 26 nM (i.e., equivalent to 0.24 and 1.7 mg·L$^{-1}$, respectively, based on

a molecular mass of laccase of 65 kDa [5]) were selected to ensure that <90% conversion of 10 µM estradiol (a fast substrate) and 10 µM cumylphenol (an intermediate rate substrate) could be achieved, respectively, in a 3-h reaction period. In the reaction matrix of CPB at pH 5, 25 °C and 1 atmosphere pressure, the saturated oxygen concentration, $[O_2]_{sat}$, at the beginning of experiments before initiating catalytic reactions was approximately 250 µM [5].

Batch reactions were conducted in 8-mL borosilicate vials maintained at 25.0 ± 0.05 °C by placing the vials in water jackets connected to a water bath. Reactions were initiated by adding an aliquot of laccase stock solution to reaction mixtures, followed by vigorous mixing for approximately 10 seconds while temporarily capped with PTFE-coated caps. Reactions were typically conducted over periods of up to 3 h without mixing, since it had been shown previously that continuous vigorous mixing can negatively impact the enzyme [5,6], and because mass-transfer limitations did not play a role in limiting the rate of reaction in the range of substrate concentrations used here [5,6]. Oxygen consumed in the catalytic reaction was replenished through exposure to ambient air, where the first-order mass transfer coefficient for oxygen diffusing from air into the batch reaction mixture under these conditions was $k_L a = 1.05 \times 10^{-2}$ min$^{-1}$ [5]. Note that, for the reactions conducted here with very low substrate concentrations, the excess of oxygen in reacting mixtures relative to substrates and its replenishment through mass transfer resulted in oxygen concentrations that were approximately constant at 250 µM over the course of experiments.

At the beginning of the reaction, 250-µL samples were withdrawn, followed by further withdrawals at regular time intervals over a typical 3-h reaction period using Hamilton glass syringes equipped with metal needles and a calibrated plunger support. Each time a sample was taken, the reaction vial contents were mixed completely. Samples were quickly transferred to 2-mL injection vials containing 4 µL of 5 N sulfuric acid, which rapidly halted the catalytic reaction by reducing the pH of the reacting mixture to 2. The pKa values of the four substrates ensured their full availability for laccase-catalyzed oxidation at the selected reaction pH of 5.0 and for sample analysis by HPLC at pH 2 [6].

### 3.4. Kinetic Modeling

A kinetic model reported earlier [5,6] was applied in the present study to predict the time course of reactions for each of the four phenolic substrates when oxidized in the absence of other substrates. This model is referred to here as the "single-substrate model". Note that reaction conditions and the range of substrate and laccase concentrations used in the present study were identical to those used to calibrate the single-substrate model [5,6] and, as a result, the calibration parameters reported earlier were used here for modeling purposes. The equations of the single-substrate model, all variable definitions, and the values of calibration parameters are summarized in Appendix A. This single-substrate model was adapted for the purposes of the present study to derive a "multi-substrate model" that describes the transient kinetics of oxidation of mixtures of substrates.

The equations of the single-substrate and multi-substrate models were solved numerically using Polymath Professional software (Version 6.10., 2004, Polymath Software, Willimantic, CT, USA). Model equations were solved with initial conditions at $t = 0$ of concentrations of total laccase ($[E_t]$), oxygen ($[O_2]_0 = [O_2]_{sat} = 250$ µM), and substrate in the reacting mixture ($[S_n]_0$), and by using the kinetic parameters arising from the calibration of the single-substrate model (see Table A1).

## 4. Conclusions

Experiments were conducted in order to study the effects of secondary substrates on the laccase-catalyzed oxidation of a target substrate and, furthermore, the kinetics of these reactions were modeled in order to describe the transient concentrations of multiple substrates as a function of time in reacting mixtures. Results indicated that the presence of secondary substrates has an impact on the conversion achieved for a target substrate, but that the magnitude of this effect depends on the rate at which the secondary substrate is oxidized, its tendency to inactivate laccase during its oxidation, and its initial concentration. Once a quantity of enzyme is selected in order to achieve a particular level

of conversion of a target within a certain timeframe, the presence of a very slow substrate will have a minor effect on a faster target substrate, except when it is present in a high concentration. Similarly, when a slow substrate is targeted for conversion, a fast substrate is rapidly oxidized in parallel and will only have a minor impact on the rate of conversion of the target substrate. However, it was found that the impacts of such secondary substrates could be very significant when they tend to inactivate laccase. In mixed reactions where more than one substrate resulted in inactivation, modeling showed that the inactivation tended to be additive; that is, the cumulative inactivation occurs at the same level as would be expected when considering the oxidation of the substrates in isolation from each other. The exception to this was reactions in which both estradiol and triclosan were present, where evidence indicated that an additional source of inactivation or inhibition occurred that could be accounted for by the two substrates alone. The multi-substrate model demonstrated its ability to accurately predict the transient concentrations of all other combinations of substrates in a reaction mixture and, in order to do so, did not require any parameter calibration beyond that done when the model was applied to single substrates. The utility of the model was demonstrated by showing how it can be applied to gain insights into the impact of various secondary substrates of differing characteristics on the oxidation of the target compound, and how it can be used to evaluate the quantities of enzyme and reaction times required to achieve various levels of conversion of a target substrate in the absence and presence of secondary substrates.

**Author Contributions:** Conceptualization, S.R. and J.A.N.; methodology, S.R. and J.A.N.; software, S.R.; validation, S.R.; formal analysis, S.R.; investigation, S.R.; resources, J.A.N.; data curation, S.R.; writing—original draft preparation, S.R.; writing—review and editing, J.A.N.; visualization, S.R.; supervision, J.A.N.; project administration, J.A.N.; funding acquisition, J.A.N.

**Funding:** This research was funded through a Discovery Grant from the Natural Sciences and Engineering Research Council of Canada and through a James McGill Chair awarded by McGill University to J.A.N.

**Conflicts of Interest:** The authors declare no conflict of interest.

## Nomenclature

| | |
|---|---|
| $\alpha$, $\alpha_n$ | Exponent of [E*] in a kinetic equation describing the rate of substrate oxidation, with possible values of $0 < \alpha \leq 1$ (dimensionless) |
| $\beta$, $\beta_n$ | Exponent of [S] in a kinetic equation describing the rate of substrate oxidation, with possible values of $\beta \geq 1$ (dimensionless) |
| *[E]* | Concentration of laccase in the native, reduced state ($\mu$M) |
| *[E*]* | Concentration of laccase in the oxidized state ($\mu$M) |
| *[E$_i$]* | Concentration of laccase in an inactivated state ($\mu$M) |
| *[E$_t$]* | Concentration of laccase added to the reaction mixture ($\mu$M) |
| $k_1$ | Apparent second-order rate constant incorporating all steps governing the reaction between laccase in the reduced state, E, and oxygen ($\mu$M$^{-1}$·min$^{-1}$) |
| $k_i$, $k_{i_n}$ | Laccase inactivation rate constant ($\mu$M$^{-0.5}$·min$^{-0.5}$) |
| $k_s$, $k_{s_n}$ | Apparent rate constant for substrate, S, reacting with oxidized laccase, E* ($\mu$M$^{1-\alpha-\beta}$·min$^{-1}$) |
| $k_L a$ | First-order mass transfer coefficient for oxygen diffusing from ambient air into the batch reaction mixture (min$^{-1}$) |
| *N* | Total number of substrates in a reaction mixture |
| *n* | Parameter subscript denoting the nth of N substrates in a mixture |
| *[O$_2$]* | Concentration of oxygen in the reaction mixture at time t ($\mu$M) |
| *[O$_2$]$_0$* | Concentration of oxygen in the reaction mixture at t = 0 ($\mu$M) |
| *[O$_2$]$_{sat}$* | Saturation concentration of the oxygen in the reaction media ($\mu$M) |
| *[S]*, *[S$_n$]* | Concentration of substrate in the reaction mixture at time t ($\mu$M) |
| *[S]$_0$*, *[S$_n$]$_0$* | Concentration of substrate in the reaction mixture at t = 0 ($\mu$M) |
| *t* | Time since initiation of reaction of substrate and laccase (min) |

## Appendix A. Model Development

*Appendix A.1. Single-Substrate Kinetic Model*

In an earlier work [5], a model was developed to predict the time course of the laccase-catalyzed oxidation of phenol at low concentrations. It was developed based on the reactions of laccase illustrated in Figure A1.

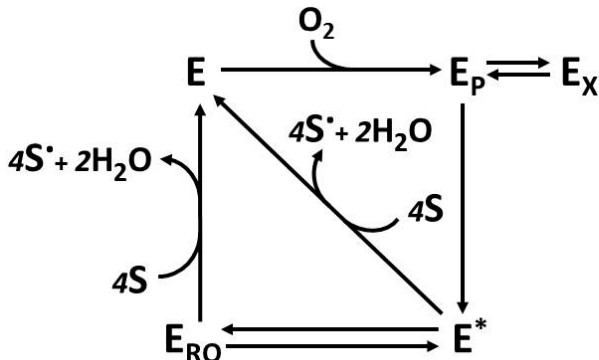

**Figure A1.** The catalytic cycle of laccase in which enzyme in the native state, E, is oxidized by aqueous oxygen, $O_2$, to produce activated enzyme, E*, which reacts with substrate, S, to produce phenoxy radicals, S·. Catalytically slow ($E_{RO}$) or non-productive ($E_P$, $E_X$) states of laccase involved in side reactions are also shown (Source: [5]).

In a subsequent work, the general applicability of the model was demonstrated by applying it to three other phenolic substrates at micromolar and sub-micromolar concentrations. For these three substrates, including estradiol, cumylphenol, and triclosan, it was necessary to incorporate an expression accounting for enzyme inactivation into the model [6], where it was assumed that inactivation results from the interaction of substrate radicals produced during the reaction, S, with the active enzyme. This model was used to predict the time course of reactions of reaction mixtures containing a single phenolic substrate and, hence, it is termed here the "single-substrate" model.

The general model consists of four differential and one mass balance equations, as follows:

$$\frac{d[O_2]}{dt} = -k_1[E][O_2] + k_L a([O_2]_{sat} - [O_2]) \tag{A1}$$

$$\frac{d[S]}{dt} = -k_s[E^*]^\alpha[S]^\beta \tag{A2}$$

$$\frac{d[E]}{dt} = -k_1[E][O_2] + k_s[E^*]^\alpha[S]^\beta \tag{A3}$$

$$\frac{d[E_i]}{dt} = k_i([E_t] - [E_i])\left(-\frac{d[S]}{dt}\right)^{0.5} \tag{A4}$$

$$[E^*] = [E_t] - [E] - [E_i] \tag{A5}$$

These equations describe the transient kinetics of laccase-catalyzed oxidation of a phenolic substrate in an aqueous reaction mixture with initial concentrations of substrate $[S]_0$, oxygen $[O_2]_0$, and enzyme $[E_t]$. Equation (A1) describes the rate of change of oxygen concentration due to its consumption in the oxidation reaction and its replenishment by mass transfer for situations where the system is open to the air. Equation (A2) describes the kinetics of phenolic substrate oxidation, and Equation (A3) describes the rate of concentration change of enzyme in the reduced form, E. The rate of formation of inactive enzyme is described by Equation (A4), where the term $[E_t] - [E_i]$ represents the total active enzyme in the mixture at any instant. Finally, the enzyme mass balance, Equation (A5), is used to

estimate the concentration of enzyme in the oxidized state, E*, at any instant. Variable definitions and their dimensions are listed in the Nomenclature above. As reported previously, when solving these equations, it can be assumed that $[O_2]_0 = [O_2]_{sat}$ for reaction mixtures that were in equilibrium with ambient air before reaction initiation at $t = 0$. Furthermore, it can be assumed that all enzyme is in the oxidized form at the start of the reaction (i.e., $[E] = 0$ and $[E^*] = [E_t]$).

As explained elsewhere [6], the model has been calibrated for single-substrate reactions of phenol, estradiol, cumylphenol, and triclosan. The kinetic parameters for each of these substrates are summarized in Table A1. The model was shown to be able to predict the time course of substrate concentrations for reactions conducted under conditions that were significantly outside of the range of its calibration. Overall, the model very accurately described the time course of reactions over several orders of magnitude of enzyme and substrate concentrations for reactions conducted over 3-h periods and longer. The kinetic parameters listed in Table A1 were used in the present study when modeling reactions of individual or mixed substrates.

**Table A1.** Modeled average rates of substrate oxidation (and relative rates with respect to phenol) during the first minute of reactions conducted at high and low enzyme concentrations at pH 5.0 and 25 °C. Also shown is the estimated percentage of enzyme inactivated over the first minute of the reaction modeled under conditions of high substrate and high enzyme concentrations where inactivation would be greatest.

| Parameters | Parameter Values Used to Model Substrate Kinetics | | | |
| --- | --- | --- | --- | --- |
| | Estradiol | Cumylphenol | Triclosan | Phenol |
| $k_L a$ (min$^{-1}$) | $1.05 \times 10^{-2}$ | | | |
| $k_1$ (µM$^{-1}$·min$^{-1}$) | $5.44 \times 10^{-1}$ | | | |
| $k_s$ (µM$^{1-\alpha-\beta}$·min$^{-1}$) | 6.75 | 0.879 | 0.271 | $4.17 \times 10^{-3}$ |
| $\alpha$ (dimensionless) | 1.00 | 0.88 | 0.87 | 0.51 |
| $\beta$ (dimensionless) | 1.18 | 1.23 | 1.39 | 1.44 |
| $k_i$ (µM$^{-0.5}$·min$^{-0.5}$) | $1.87 \times 10^{-2}$ | $5.11 \times 10^{-2}$ | $2.99 \times 10^{-2}$ | 0 |

*Appendix A.2. Multi-Substrate Kinetic Model*

The single-substrate kinetic model presented above was adapted to predict the time course of reactions of multiple substrates in a reaction mixture, as will be described below. Note that Equations (A1) and (A5) do not require adaptation because they are substrate-independent.

For a mixture of multiple substrates, equations must be written and simultaneously solved for each substrate to evaluate their respective rates of concentration change with time. That is, for each individual substrate, $n$, in a mixture of $N$ substrates, Equation (A2) can be adapted as follows to calculate the rate of concentration change of substrate $n$, $[S_n]$:

$$\frac{d[S_n]}{dt} = -k_{s_n}[E^*]^{\alpha_n}[S_n]^{\beta_n} \tag{A6}$$

Therefore, for a mixture of $N$ substrates, there will be $N$ independent equations where each substrate, $S_n$ (i.e., where $n = 1$ to $N$), will have its own rate equation and kinetic parameters, $k_{s_n}$, $\alpha_n$, and $\beta_n$.

Equation (A3) can then be adapted to account for the formation of reduced enzyme, E, in an additive manner through the parallel reactions of E* with each substrate, $S_n$, as follows:

$$\frac{d[E]}{dt} = -k_1[E][O_2] + \sum_{n=1}^{N} k_{s_n}[E^*]^{\alpha_n}[S_n]^{\beta_n} \tag{A7}$$

Similarly, the rate of enzyme inactivation in a mixture can be calculated as the sum of the rates of inactivation arising from the oxidation of each substrate. In doing this, it is assumed that the free-radical products, $S_n$, arising from the reactions of the different substrates, $S_n$, do not interact with each other in such a way that inactivation is reduced (through radical–radical reactions) or increased (through product formation). The validity of this assumption is tested in the present work. The rate of change of concentration of inactive enzyme, $[E_i]$, may be expressed as

$$\frac{d[E_i]}{dt} = ([E_t] - [E_i]) \sum_{n=1}^{N} k_{i_n} \left( -\frac{d[S_n]}{dt} \right)^{0.5} \tag{A8}$$

Therefore, the kinetic model for the laccase-catalyzed oxidation of a mixture of $N$ substrates consists of Equations (A1), (A5), (A7), and (A8), along with Equation (A6) adapted $N$ times for the $N$ substrates. The initial conditions for the numerical solution of model equations are $[O_2] = [O_2]_0$, $[S_n] = [S_n]_0$, $[E^*] = [E_t]$, $[E] = 0$, and $[E_i]_0 = 0$ at $t = 0$.

For the purposes of this study, the model coefficients used included the kinetic constants listed in Table A1 for the four phenolic substrates [6], an oxygen mass-transfer coefficient, $k_L a$, of $1.05 \times 10^{-2}$ min$^{-1}$, and $[O_2]_0$ equal to the saturation concentration, $[O_2]_{sat}$, of 250 μM [5]. The methods used to calibrate these parameters are reported elsewhere [5,6].

## Appendix B. Classification of Substrates

For the purposes of this study, substrates were classified as either fast, intermediate, or slow with respect to their relative rates of oxidation. Note that, due to the non-linear nature of Equation (2), it was not possible to directly compare the kinetic parameters shown in Table A1 in order to carry out such a classification. Instead, they were classified based on estimates of the rate of substrate oxidation, $\frac{d[S]}{dt}$ (μM·min$^{-1}$), and modeled under fixed reaction conditions of high and low substrate and enzyme concentrations. Thus, the model equations summarized in Section A1 were solved using the initial conditions listed in the first column of Table A2 in order to determine the average reaction rate in the first minute of the reaction. Note that these modeled initial conditions of substrate and enzyme concentrations were the same as those used in kinetic experiments in the present study (i.e., see Figures 4–6 in the main body).

The calculated average reaction rates were tabulated for all four substrates in Table A2. For ease of comparison, all rates in Table A2 were also expressed relative to the rates of the slowest substrate, phenol. Over the range of conditions modeled, it can be seen that the rate of oxidation of estradiol was between 33 and 235 times faster than phenol under the same conditions. Cumylphenol was the next fastest substrate, with relative oxidation rates between 14 and 53 times that of phenol. Triclosan had relative rates between 7.1 and 17 times faster than phenol, and was between 2 and 3 times slower than cumylphenol. For these reasons, estradiol was deemed a "fast" substrate, cumylphenol and triclosan were deemed "intermediate" substrates, and phenol was deemed a "slow" substrate.

Similarly, for the purposes of this study, the substrates were also classified in terms of their tendency to inactivate laccase during the catalytic reaction. In order to classify them, the model was used to predict the quantity of inactivation that would occur during the first minute of reaction under a condition of high enzyme and substrate concentrations where the rate of inactivation would be greatest [6]. As can be seen in the bottom rows of Table A2, with $[S]_0 = 10$ μM and $[E_t] = 26.0$ nM, it was predicted that 3.6%, 2.2%, 1.5%, and 0% of the enzyme supplied to the reaction would be inactivated in the first minute when oxidizing cumylphenol, estradiol, triclosan, and phenol, respectively. As such, in terms of their relative tendency to inactivate laccase during the catalytic reaction, cumylphenol was deemed to be "high", estradiol and triclosan were deemed to be "moderate", and phenol was deemed to be "non-inactivating". Note that phenol has been observed to inactivate laccase, but only at elevated concentrations in excess of 500 μM [14]. As such, for the purpose of this study, where very low concentrations of phenol were used, this substrate is considered to be "non-inactivating".

**Table A2.** Modeled average rates of substrate oxidation (and relative rates with respect to phenol) during the first minute of reactions conducted at high and low enzyme concentrations at pH 5.0 and 25 °C. Also shown is the estimated percentage of enzyme inactivated over the first minute of the reaction modeled under conditions of high substrate and high enzyme concentrations where inactivation would be greatest.

| Reaction Conditions or Classification | Substrate | | | |
|---|---|---|---|---|
| | Estradiol | Cumylphenol | Triclosan | Phenol |
| | Rate of Substrate Oxidation ($\mu$M·min$^{-1}$) (Relative Rate with Respect to Phenol; Dimensionless) | | | |
| $[S]_0 = 1\ \mu M$, $[E_t] = 3.64$ nM | $2.31 \times 10^{-2}$ (85) | $6.17 \times 10^{-3}$ (23) | $2.04 \times 10^{-3}$ (7.5) | $2.73 \times 10^{-4}$ (1) |
| $[S]_0 = 1\ \mu M$, $[E_t] = 26.0$ nM | $1.52 \times 10^{-1}$ (235) | $3.42 \times 10^{-2}$ (53) | $1.12 \times 10^{-2}$ (17) | $6.48 \times 10^{-4}$ (1) |
| $[S]_0 = 10\ \mu M$, $[E_t] = 3.64$ nM | $2.11 \times 10^{-1}$ (33) | $8.85 \times 10^{-2}$ (14) | $4.59 \times 10^{-2}$ (7.1) | $6.50 \times 10^{-3}$ (1) |
| $[S]_0 = 10\ \mu M$, $[E_t] = 26.0$ nM | 1.43 (80) | $5.01 \times 10^{-1}$ (28) | $2.54 \times 10^{-1}$ (14) | $1.78 \times 10^{-2}$ (1) |
| Classification according to relative rate of oxidation | Fast | Intermediate | Intermediate | Slow |
| | Laccase inactivation (%) | | | |
| $[S]_0 = 10\ \mu M$, $[E_t] = 26.0$ nM | 2.2 | 3.6 | 1.5 | 0 |
| Classification according to relative tendency to inactivate | Moderate | High | Moderate | Non-Inactivating |

## Appendix C. Investigation of Impacts of Secondary Substrates Using the Multi-Substrate Model

The multi-substrate kinetic model can be used to gain insights into the impacts of secondary substrates on the rate of conversion of a target substrate, especially under conditions that may be difficult to explore experimentally. For example, simulations were conducted here to investigate the impacts of hypothetical secondary substrates on the conversion of cumylphenol (intermediate rate, highly inactivating) and estradiol (fast substrate, moderately inactivating) as target substrates. In these simulations, the impacts of hypothetical substrates, denoted as X, that were kinetically similar to the target substrates, were investigated. This was done by setting the kinetic parameters of the hypothetical secondary substrate (i.e., $k_{S_X}$, $\alpha_X$ and $\beta_X$) equal to those of the target substrates (see Table A1). The effect of the concentration of this hypothetical secondary substrate was then investigated by varying its initial concentration, $[S_X]_0$, in simulations. Furthermore, the effect of the tendency of the hypothetical secondary substrate to inactivate the enzyme was investigated by varying its inactivation rate constant, $k_{i_X}$.

First, in order to evaluate the impact of increasing the concentration of kinetically similar but non-inactivating substrates, simulations were performed in which the initial concentration of the secondary substrate, X, was varied, but where its inactivation rate constant, $k_{i_X}$, was set to 0. The results of simulations are shown in Figures A2a and A3a for cumylphenol and estradiol, respectively, where the initial concentration of the target substrate was equal to 10 $\mu$M, and where the effect of the initial concentration of the hypothetical secondary substrate was simulated at 0, 10, and 20 $\mu$M.

The corresponding predicted distributions of enzyme over time for these simulations are shown in Figures A2b and A3b. As can be seen in Figure A2a, the impact of a kinetically similar but non-inactivating substrate on the predicted rate of oxidation of cumylphenol (intermediate rate, highly inactivating) is very little. In contrast, similar simulations using estradiol as the target substrate indicate that a non-inactivating but kinetically similar secondary substrate can have a significant impact on the rate of oxidation of the target substrate, and that this impact increases with concentration, as shown in Figure A3a.

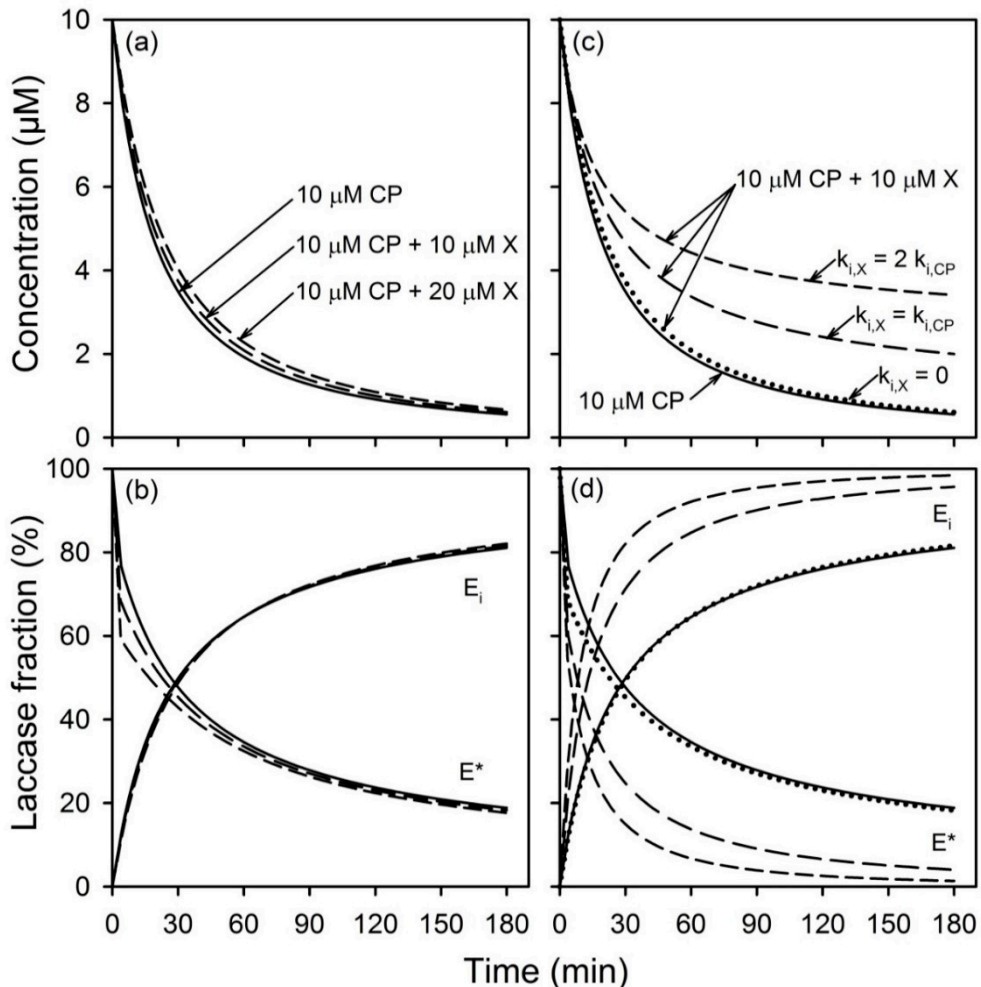

**Figure A2.** Predicted impacts of a hypothetical secondary substrate, X, on the kinetics of oxidation of the target substrate, cumylphenol (CP), and the corresponding fractions of laccase in the oxidized, E*, and inactivated, E$_i$, states over time: (**a**,**b**) effect of varying the initial concentration, $[S_X]_0$, of the secondary substrate, X, when it is non-inactivating (i.e., $k_{i_X}$ = 0); (**c**,**d**) effect of varying the inactivation rate constant, $k_{i_X}$, of the secondary substrate, X, for a fixed initial secondary substrate concentration, $[S_X]_0$, of 10 μM. For all simulations, the initial concentration of the target substrate was 10 μM and the total enzyme supplied to the reaction, $[E_t]$, was 26 nM. The fractions of enzyme in the E* and E$_i$ states over time are expressed as a percentage of the total enzyme, $[E_t]$. The fraction of enzyme in the native state, E, is not shown.

As can be seen in Figures A2b and A3b, the presence of the non-inactivating secondary substrates is predicted to have only a minor impact on the rate of accumulation of inactive enzyme over time, even when present in significant concentration. However, these figures show that such secondary substrates significantly impact on the fraction of enzyme in the oxidized state, E*, especially in the early stages of the reaction. Furthermore, as shown in Figure A3b, the impact on the enzyme distribution is especially significant when the target and secondary substrates are both kinetically very fast and particularly when the secondary substrate is at a relatively high concentration with respect to the target substrate.

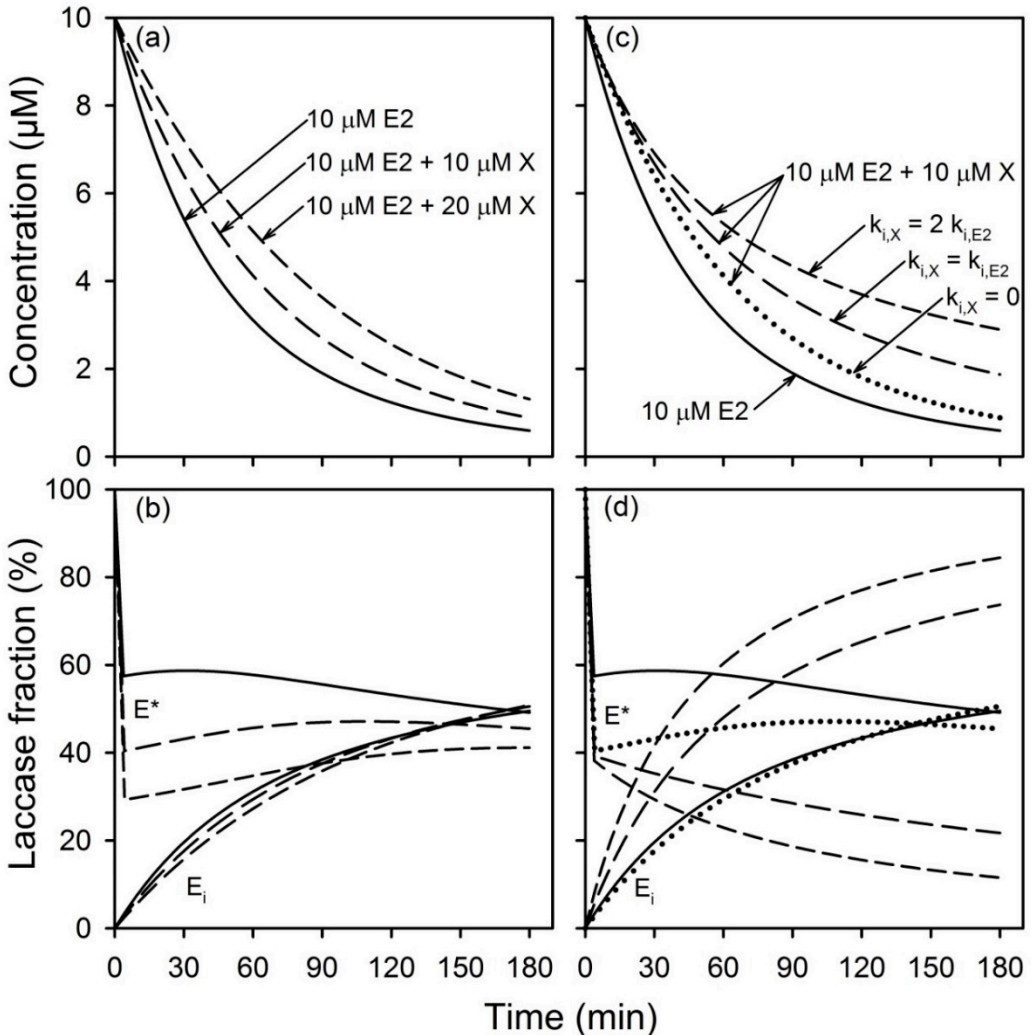

**Figure A3.** Predicted impacts of a hypothetical secondary substrate, X, on the kinetics of oxidation of the target substrate, estradiol (E2), and the corresponding fractions of laccase in the oxidized, E*, and inactivated, $E_i$, states over time: (**a,b**) effect of varying the initial concentration, $[S_X]_0$, of the secondary substrate, X, when it is non-inactivating (i.e., $k_{i_X} = 0$); (**c,d**) effect of varying the inactivation rate constant, $k_{i_X}$, of the secondary substrate, X, for a fixed initial secondary substrate concentration, $[S_X]_0$, of 10 µM. For all simulations, the initial concentration of the target substrate was 10 µM and the total enzyme supplied to the reaction, $[E_t]$, was 3.64 nM. The fractions of enzyme in the E*and $E_i$ states over time are expressed as a percentage of the total enzyme, $[E_t]$. The fraction of enzyme in the native state, E, is not shown.

In a second set of simulations, the impact of secondary substrates that are kinetically similar to the target substrate but with different tendencies to inactivate the enzyme were evaluated. The results are shown in Figures A2c and A3c. Simulations are shown for reactions of the 10-µM target substrate alone and in the presence of 10 µM of a hypothetical secondary substrate that is non-inactivating ($k_{i_X} = 0$) or that have the same or twice the tendency to inactivate the enzyme as compared to the target substrate. The predicted distributions of enzyme over time for these simulations are shown in Figures A2d and A3d. From these simulations, it is clear that the impact of an inactivating secondary substrate on the rate of oxidation of the target substrate is very significant. Furthermore, the magnitude of this impact is important for both intermediate and fast substrates. This is further demonstrated by the significant impact that the inactivating substrate has on the quantity of inactive enzyme, $E_i$, accumulated over time, and a corresponding reduction in the amount of enzyme in the E* state that is available to oxidize

the substrates. The overall effect of this simultaneous accumulation of $E_i$ and depletion of $E^*$ is a significant negative impact on the rate of oxidation of the target substrate. Similar simulations will show that this negative impact worsens with an increasing concentration of the secondary substrate (not shown).

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
