# Peer review of "Laccase-Catalyzed Oxidation of Mixed Aqueous Phenolic Substrates at Low Concentrations"

_catalysts, doi:10.3390/catal9040368_

Round 1

Reviewer 1 Report

In the paper, the authors built a kinetic model to predict phenols oxidation by using laccase enzyme. The effect of secondary materials on the oxidation rate is studied. Before publication, the following issues the authors may consider improving the quality of this manuscript. 1. The author chose 4 phenols: phenol, estradiol, cumylphenol, and triclosan and built the model. Does this model also apply to other phenolic substrates (e.g. cresol, bisphenol A)? The author can use other material to verify accuracy. 2. Does this model also apply in other pH value? 3. The authors studied the system at low concentration. Is it reliable under normal or high concentration?

Author Response

Response to Reviewer 1

Reviewer’s Comments

In the paper, the authors built a kinetic model to predict phenols oxidation by using laccase enzyme. The effect of secondary materials on the oxidation rate is studied. Before publication, the following issues the authors may consider improving the quality of this manuscript. 1. The author chose 4 phenols: phenol, estradiol, cumylphenol, and triclosan and built the model. Does this model also apply to other phenolic substrates (e.g. cresol, bisphenol A)? The author can use other material to verify accuracy. 2. Does this model also apply in other pH value? 3. The authors studied the system at low concentration. Is it reliable under normal or high concentration?

Authors’ Response

We thank Reviewer 1 for taking the time to conduct this review and provide us with constructive
feedback. Our responses to the numbered comments provided by the reviewer are as follows:

1. As we describe in this work, this semi‐empirical model was initially derived for phenol (see [5]), which is a relatively slow substrate. In a follow‐up paper cited in the present manuscript (see [6]), we demonstrated that this model could be used for other substrates (including estradiol,
cumylphenol, and triclosan), each of which are important pollutants and, moreover, are
characterized by very different rates of reaction and tendencies to inactivate the enzyme. The model was able to describe the kinetics of the reactions of all 4 substrates over orders of magnitude of enzyme and substrate concentration and was able to predict the kinetics of reactions outside of the enzyme and substrate concentration ranges used for model calibration. As such, we are confident that the model can accurately describe the kinetics of reactions of many phenols. While cresol and bisphenol A are substrates of the enzyme and are important pollutants, we are of the view that the objective of this paper was not to come up with an exhaustive list of substrates whose kinetics could be modelled using this model. Rather, the intent of this paper was to explore the effects of mixed substrates on the reactions of the individual substrates and to apply the model to gain insights into what happens with such mixtures. We leave it to other researchers to pursue the application of the model to other substrates.

2. We have applied this model at a pH that is in the vicinity of the typical conditions at which
wastewaters would be treated and, moreover, at a pH that reflects the optimum activity of the
enzyme. It was beyond the scope of the present study to model the reactions of mixtures under other conditions (e.g., temperature, pH, presence of dissolved compounds or suspended solids, etc.). The effect of these and many other variables can certainly be the subject of future investigations.

3. As we point out in our previous work cited in this manuscript (see [5, 6]), this model was specifically developed to model the unusual kinetics that occur when targeting very low concentration of substrates for oxidation. Under these conditions, certain stages of the reaction cycle become particularly rate limiting, which is quite different from what is experienced with reactions at high substrate and enzyme concentrations. For examples of what happens at much higher concentrations, particularly with respect to the stoichiometry between oxygen and phenolic substrate and inactivation, please see previous work which is cited in our article (see [14]).

Reviewer 2 Report

This paper reported a model describing the oxidation of phenolic substrates at low concentrations catalyzed by laccase. The model can well describe the competitive adsorption of other species and the deactivation of the laccase by products or byproducts. Whereas the authors have attempted to address an important research question in kinetics of biocatalysis, there are several issues that need to be addressed before the paper is suitable for publication on Catalysts, as listed below:

1.        On lines 68 – 80, the author introduced that other substrate can influence the oxidation by 1) “First, the rate of oxidation of the target substrate may be hindered by competition between it and secondary substrates for the enzyme”; 2) “Secondly, a fast secondary substrate could be preferentially oxidized by  the enzyme, resulting in a sequential delay in the oxidation of the slower target substrate until much of the secondary substrate is depleted.”, and other factors. Factor 1 and factor 2 are essentially the same and should not be separated, both of which originates from the competitive adsorption.

2.       On line 71, the author mentioned “In this instance, it is hypothesized that slower secondary substrates that may tend to occupy the active site for longer periods might have a greater negative impact on the rate of oxidation of the target substrate than faster secondary substrates.” This hypothesis is not complete. When species B affects the reaction rate of species A by competitive adsorption, only the adsorption coeffect of B matters but not the reaction rate of B. It is possible that the coverage of adsorbed species B is very small but the reaction rate of B is very fast, in which case B still has negligible effect on reaction rate of A.

3.       In Figure 1, the authors showed that 1 – 50 μM phenol negligibly affect the conversion of E2 (above 90% in all case), based on which the conclusion was made that phenol did not competitively occupy the catalytic sites of enzyme. It is notable that when the conversion is above 90%, it is likely in the kinetically-insensitive regime. To test whether there is any competitive adsorption, conversion with or without additives should be measured and compared in the region of less than 10% (or 20%).

4.       The proposed reaction model contains the mass transfer term of O2 (equation 2, line 370). Based on the fitting results, is there any mass transfer effect of O2 on the reaction rate? It will be also more informative to add the concentration of O2 at different time under various conditions in the manuscript.

5.       In equation 2, line 370, the term k1*[E]*[O2] indicates the reaction of O2 with the reduced enzyme is 1st order with respect to the O2 concentration. This assumption needs to be justified.

6.       In the model, the oxidation rate of species n is described as k[E*]α[Sn]β and the α and β can be further fitted based on the experimental data. Whereas the empirical form can well describe the reaction results, it would be more ideal to us a more detailed and informative kinetic model. For example, if assuming the equilibrated adsorption of various species (with Kn as adsorption coefficient of species n) and kn as the rate constant of adsorbed species n, the oxidation rate of species n will be kn[E*]Kn[Sn]/                                               (Ki[Si]). This form makes more kinetic sense and also decrease the variables to be fitted.

Author Response

Response to Reviewer 3

Reviewers’ Comments

The manuscript describes the results of experiments conducted to evaluate effects on Trametes
versicolor laccase arising from the simultaneous presence of mixed phenolic substrates at low
concentrations during the wastewater detoxification. Phenol, estradiol, cumylphenol and triclosan substrates were selected and their oxidation rates and tendencies to inactivate laccase were studied. Each of these substrates represents diverse groups of contaminants of environmental importance. A multi‐substrate kinetic model was used to predict the transient kinetics of reactions of mixtures and the model predictions were compared with the experimental results. The utility of the model was demonstrated for gaining insights into reaction phenomenon and for evaluating the feasibility of oxidizing targeted substrates in the presence of other substrates. The article was well written with sufficient background information and clear presentation of results and conclusions. The results have great significance with scientific soundness. The manuscript can be accepted in its current form for publication.

However, I would like to suggest the authors include the structures of the four substrates to demonstrate the diversity, and possible early‐stage products generated during the oxidation by laccase which is missing in my opinion.

Authors’ Response

We thank Reviewer 3 for this review. We are very appreciative of the opinions expressed by the
reviewer and particularly grateful for the comment about the significance of the results and the
scientific soundness. This was exactly what we hoped to achieve in this complex study, which took many years to conduct.

With respect to the suggested inclusion of the structures of the parent phenols and the early‐stage products, firstly we felt that the structures of the parent compounds are widely known and available through typical sources. We do not believe that, in what is already a fairly long paper with many figures, the inclusion of these structures would add much significant information and, therefore, their inclusion is not warranted.

Secondly, with respect to the inclusion of structures of possible early‐stage products of the oxidation, we do not feel that it would be appropriate to present such structures for several reasons. Firstly, the number of combinations of structures that are possible in mixtures of multiple reacting substrates are huge. For example, the dimeric products initially arising from the oxidation of 4 compounds in a mixture would include 16 different combinations of parent compounds, each with multiple possible structures.There are far too many too include in this paper. Secondly, based on previously‐published work by others, we know that the production of trimers is also possible (i.e., 64 possible trimer combinations can arise from the simultaneous oxidation of 4 substrates, each with many conformations). And, finally, given that we did not characterize the products of our reactions, we think that it would be too hypothetical to present such possible structures in this paper.

Reviewer 3 Report

The manuscript describes the results of experiments conducted to evaluate effects on Trametes versicolor laccase arising from the simultaneous presence of mixed phenolic substrates at low concentrations during the wastewater detoxification. Phenol, estradiol, cumylphenol and triclosan substrates were selected and their oxidation rates and tendencies to inactivate laccase were studied. Each of these substrates represents diverse groups of contaminants of environmental importance. A multi-substrate kinetic model was used to predict the transient kinetics of reactions of mixtures and the model predictions were compared with the experimental results. The utility of the model was demonstrated for gaining insights into reaction phenomenon and for evaluating the feasibility of oxidizing targeted substrates in the presence of other substrates. The article was well written with sufficient background information and clear presentation of results and conclusions. The results have great significance with scientific soundness. The manuscript can be accepted in its current form for publication. 

However, I would like to suggest the authors include the structures of the four substrates to demonstrate the diversity, and possible early-stage products generated during the oxidation by laccase which is missing in my opinion.       

Author Response

Response to Reviewer 2

Reviewers’ Comments

This paper reported a model describing the oxidation of phenolic substrates at low concentrations catalyzed by laccase. The model can well describe the competitive adsorption of other species and the deactivation of the laccase by products or byproducts. Whereas the authors have attempted to address an important research question in kinetics of biocatalysis, there are several issues that need to be addressed before the paper is suitable for publication on Catalysts, as listed below:

1. On lines 68 – 80, the author introduced that other substrate can influence the oxidation by 1) “First, the rate of oxidation of the target substrate may be hindered by competition between it and secondary substrates for the enzyme”; 2) “Secondly, a fast secondary substrate could be preferentially oxidized by the enzyme, resulting in a sequential delay in the oxidation of the slower target substrate until much of the secondary substrate is depleted.”, and other factors. Factor 1 and factor 2 are essentially the same and should not be separated, both of which originates from the competitive adsorption.

2. On line 71, the author mentioned “In this instance, it is hypothesized that slower secondary substrates that may tend to occupy the active site for longer periods might have a greater negative impact on the rate of oxidation of the target substrate than faster secondary substrates.” This hypothesis is not complete. When species B affects the reaction rate of species A by competitive adsorption, only the adsorption coeffect of B matters but not the reaction rate of B. It is possible that the coverage of adsorbed species B is very small but the reaction rate of B is very fast, in which case B still has negligible effect on reaction rate of A.

3. In Figure 1, the authors showed that 1 – 50 μM phenol negligibly affect the conversion of E2 (above 90% in all case), based on which the conclusion was made that phenol did not competitively occupy the catalytic sites of enzyme. It is notable that when the conversion is above 90%, it is likely in the kinetically insensitive regime. To test whether there is any competitive adsorption, conversion with or without additives should be measured and compared in the region of less than 10% (or 20%).

4. The proposed reaction model contains the mass transfer term of O2 (equation 2, line 370). Based on the fitting results, is there any mass transfer effect of O2 on the reaction rate? It will be also more informative to add the concentration of O2 at different time under various conditions in the manuscript.

5. In equation 2, line 370, the term k1*[E]*[O2] indicates the reaction of O2 with the reduced enzyme is 1st order with respect to the O2 concentration. This assumption needs to be justified.

6. In the model, the oxidation rate of species n is described as k[E*]α[Sn]β and the α and β can be further fitted based on the experimental data. Whereas the empirical form can well describe the reaction results, it would be more ideal to us a more detailed and informative kinetic model. For example, if assuming the equilibrated adsorption of various species (with Kn as adsorption coefficient of species n) and kn as the rate constant of adsorbed species n, the oxidation rate of species n will be kn[E*]Kn[Sn]/(Ki[Si]). This form
makes more kinetic sense and also decrease the variables to be fitted.

Authors’ Response

We thank Reviewer 2 for this review and for the time and thought put into it. Our responses to the
individual numbered comments are itemized below:

1. We agree with the Reviewer’s suggestion regarding the suggested modifications to the
manuscript. The Reviewer is correct that the two factors that were presented are essentially
different facets (one related to slow secondary substrates and the other related to fast
secondary substrates). Given this, we have modified the text accordingly.

2. With respect this comment, it appears that the Reviewer doesn’t differentiate between the first step of adsorption into the active site of the enzyme and the next step that involves its
oxidation. Both steps can impose their own rate limitations on the catalytic process and, as such, we see these as two separate processes, even though they are difficult to differentiate when studying a reacting system. Unfortunately, we do not understand what is meant by the “coverage” issue raised by the reviewer. Could it be that the Reviewer is referring to a solid catalyst in which surface adsorption on many possible active sites is an issue? This is not the case in an enzymatic system where the enzyme has a single active site to which a substrate must bind. Given this, we are unable to address this comment.

3. For several reasons, we don’t agree that under these conditions of estradiol concentration and conversion that this system is likely in a “kinetically‐insensitive regime” and that this might influence our findings. First, we have shown reactions in this paper and earlier work cited here that we have achieved much greater than 90% conversion of estradiol at this same initial concentration and also at lower concentrations. Secondly, we purposely limited the quantity of enzyme used in the reactions shown in Figure 1 to slow the reaction and limit the conversion to a level that was sufficiently below 100% where effects would be evident. Under these conditions, the cumulative impact of phenol over the course of the 3‐hour reaction time would be evident, especially with the very high phenol concentrations used. Finally, it is our interest in the experiments shown in Figure 1 to demonstrate that if you wish to achieve a high level of conversion of estradiol, even with a low quantity of enzyme, it can be done in the presence of very high concentrations of phenol. This, in fact, is the entire point of this study. If we had limited the experiment to achieve a lower conversion of estradiol in the 3‐hour reaction time (say, 50%), then it would not lead us to the conclusion that we are looking for; i.e., a fast substrate like estradiol, even when present in low concentrations, can be oxidized by a very low concentration of enzyme in the presence of massive amounts of phenol (including 500 μM phenol, as mentioned in the text after Figure 1). Our conclusions are further reinforced by Figure 2, which shows the time course of reactions of estradiol in the presence of phenol and where we achieve a much higher level of conversion than 90%. Finally, with respect to the last suggestion of the reviewer, it is not clear to us which additives (nor how such additives) would serve the purpose intended by the reviewer.

4. As we mention in the paper (see Materials and Methods section) “Reactions were typically
conducted over periods of up to 3 hours without mixing, since it had been shown previously that continuous vigorous mixing can negatively impact the enzyme [5, 6] and because mass transfer limitations did not play a role in limiting the rate of reaction in the range of substrate
concentrations used here [5, 6].” As explained in these previous studies, when the initial
concentration of oxygen is 250 μM and the amount of substrate being converted is in the range of 1 to 12 μM, the depletion of oxygen over the course of experiments is very low and is, in fact, insignificant, given its continuous replenishment through mixing. This is what is expected when applying the enzyme to the treatment of wastewaters with only trace level concentrations of target substrates. However, in order to address the concern of the Reviewer, we have inserted the following sentence into the Materials and Methods section to make this clear: “Note that, for the reactions conducted here with low substrate concentrations, the high excess of oxygen in reacting mixtures relative to substrates and its replenishment through mass transfer resulted in oxygen concentrations that were approximately constant at 250 μM over the course of experiments.”

5. In our prior work, we explained the development of the model in great detail (see [5]). This
model was developed on using the Law of Mass Action which is widely used in modelling
reaction kinetics (https://en.wikipedia.org/wiki/Law_of_mass_action) based on the proposition (dating from the 19th century) that the rate of a chemical reaction is directly proportional to the product of the concentrations of the reactants. This is the most fundamental starting point of any modelling exercise that attempts to describe the kinetics of reactions of species in a mixture, which in this case the rate of change of oxygen arising from the oxidation of reduced enzyme by oxygen. The Law of Mass Action is the starting point for the derivation of all enzymatic models, whether it be the Michaelis‐Menten model
(https://en.wikipedia.org/wiki/Michaelis%E2%80%93Menten_kinetics) or others.

6. We presented the detailed derivation of our single‐substrate model in our previous work (see [5]), which is based on a semi‐empirical approach that accounts for the various stages of the reaction in the catalytic cycle and for the mechanism of free‐radical inactivation. We also showed in [5] that it is impractical to develop a purely mechanistic model to account for the detailed steps of the reaction (especially those that will become relevant at very low substrate concentrations) because this would require measurements of species in the reaction that cannot be measured in real‐time using current technologies. Moreover, the use of such an approach result in a model that would be impossible to calibrate due to the number of kinetic coefficients required. This would be an impractical model that nobody could apply. As such, in our work, we took a semi‐empirical approach to modelling where we demonstrated that the model can predict the time course of reactions of 4 very different substrates over many orders of magnitude of enzyme and substrate concentrations. We are not aware of any model in the literature that has this capability. In contrast, the reviewer has suggested that we use an equilibrium modeling approach (i.e., that produces the expression he suggests above) which he/she contends would make more “kinetic sense”. We fundamentally disagree with this suggestion. The approach suggested by the Reviewer is often used by researchers to describe the steady‐state kinetics of enzymatic reactions in which inhibition (not inactivation) is observed. We wish to note that we have not encountered any studies where such an approach resulted in a kinetic model that could accurately describe the time‐course of reactions, nor have any predictive capability. Moreover, the basis for the development of the expression suggested by the reviewer does not reflect the mechanism nor inactivation occurring in the laccase system. Thereover, it would not be mechanistically correct to use this expression in our model. And,
finally, we are confident that a model based on such an approach would not have the predictive abilities that we have demonstrated in this paper and in our previous work (see [5, 6]).

Reviewer 4 Report

The manuscript entitled “Laccase-Catalyzed Oxidation of Mixed Aqueous Phenolic Substrates at Low Concentrations” investigates the...

Introduction is well written, giving the lack of literature and relevant published studies. Materials and methods are appropriately described. Results are also described in details, but the discussion is poor. Please add more discussion. In general, the structure of manuscript is good, but it needs minor editing as some sentences are too long.

Some specific comments:

Abstract should be rewritten in order to stand alone, for example in lines 15-16 “Slower and faster substrates” should be mentioned.

Lines 31-32: please add the values of concentrations in the following statements “… relatively high substrate concentrations…” and “… greater than those encountered in real wastewaters.”

Lines 50-53: please rewrite the sentence “The utility of the model…”. It is too long. There are also many other too long sentences. Please check them throughout the manuscript.

Lines 99-106: This paragraph should be incorporated in the previous one, which describes the objectives of the study. Any literature (in particular the references 5, 6, 13 and 14) concerning enzyme and substrates should be given before the objectives. Do not mix the literature with the objectives of the study. So, please rearrange the two last paragraphs.

Discussion should be enriched. Compared your results with other similar studies. Also it is suggested  to add a short discussion about the enzyme deactivation and how substrates affect enzyme activity.

Author Response

Response to Reviewer 4

Reviewers’ Comments

Introduction is well written, giving the lack of literature and relevant published studies. Materials and methods are appropriately described. Results are also described in details, but the discussion is poor. Please add more discussion. In general, the structure of manuscript is good, but it needs minor editing as some sentences are too long.

Some specific comments:

1. Abstract should be rewritten in order to stand alone, for example in lines 15‐16 “Slower and
faster substrates” should be mentioned.

2. Lines 31‐32: please add the values of concentrations in the following statements “… relatively
high substrate concentrations…” and “… greater than those encountered in real wastewaters.”

3. Lines 50‐53: please rewrite the sentence “The utility of the model…”. It is too long. There are also many other too long sentences. Please check them throughout the manuscript.

4. Lines 99‐106: This paragraph should be incorporated in the previous one, which describes the objectives of the study. Any literature (in particular the references 5, 6, 13 and 14) concerning enzyme and substrates should be given before the objectives. Do not mix the literature with the objectives of the study. So, please rearrange the two last paragraphs. Discussion should be enriched. Compared your results with other similar studies. Also it is suggested to add a short discussion about the enzyme deactivation and how substrates affect enzyme activity.

Authors’ Response

We thank Reviewer 4 for this review and for taking the time to prepare it. Our responses to the general comments of the reviewer and those that we have numbered above for ease of reference are provided below.

In response to the general comments made, we note that Reviewer 4 expressed the opinion that “the discussion is poor,” without providing any observations or details on why it is poor or any constructive suggestions for its improvement. In our opinion, the discussion distributed throughout the paper is already very substantial. That is, we address all hypotheses raised in the paper, show how we have achieved the stated objectives, and explore the consequences of the findings through case studies in which the model is used to explore the feasibility of oxidizing substrates at low concentration in mixtures. As such, we have not made any significant changes to the discussion content of the paper in response to this Reviewer’s suggestion.

1. In fact, we have already written the abstract to stand alone to present the reasons for the work, the objectives and scope, and the general conclusions. Details are also provided about which substrates were included in the study. Given the word limit imposed on abstracts, it proves impossible to add further details without sacrificing important content in the abstract, especially about the general findings of the study. With respect to the specific suggestion regarding “slower and faster substrates”, these are observation relevant to all substrates studied (i.e., they include those that are slow and those that are fast) and, as such, it is unnecessary to repeat the list of substrates here, which was included in the previous sentence.

2. We have cited 4 references in the sentence preceding the one on lines 31‐32 (original manuscript numbering) to reflect that fact that prior studies were conducted in concentration ranges that were very high relative to what is found in actual wastewaters. In each of these papers, a number of substrates were studied, each with their own initial concentrations that were substantially different than concentrations typically reported in wastewaters and that were frequently cited in these papers. If we modified the sentence to address the concern of the Reviewer, it would be too cumbersome with the detailed information of limited value. Instead, we assume that the reader can retrieve these cited papers and confirm that the concentrations used were typically much greater than what are typically observed in wastewaters. However, in light of the Reviewer’s comment, this, we have slightly modified the text to indicate that the studies themselves report the use of initial substrate concentrations that are often higher than measured in real wastewaters.

3. While we have shortened the sentence by breaking it into two, in fact there is was nothing incorrect with the original sentence, both grammatically or stylistically. However, we have reviewed the entire paper to ensure that other sentences are not too long and have made adjustments here and there.

4. While we understand that this Reviewer has stylistic preferences, we disagree with the suggestion made. That is, it is not uncommon for the objective of a study to be stated and then the scope of the investigation be defined thereafter. The scope defines the limits/boundaries of the conditions under which the objective will be pursued. In this case, the paragraph that follows the objectives first defines the scope of the research by identifying the enzyme that was the subject of this study, with supporting references that justify this selection. Then this paragraph identifies the 4 substrates selected to be part of the study, with references that justify the environmental importance of these substrates. We also indicate that these were also selected because their oxidation by laccase had been modelled previously, which was essential for the purposes of this study. We should note that the writing of an objective followed by the description of the scope of the study is not an
uncommon style for the writing of manuscripts of papers and doctoral or masters theses. In fact, when we train our graduate students at our institution regarding the preparation of manuscripts, this is part of their education.

With respect to the suggestion that we enrich the discussion, as mentioned above, it is not clear how it needs to be enriched. The paper already includes substantial discussion that ties together all of the findings reported and, moreover, demonstrates the implications of the work including examples of the application of the model. We do not think that additional discussion is warranted, especially if such added discussion were to become too hypothetical and unsupported by the results of the study. Also, with respect to the Reviewer’s request to compare our results to previous studies, we are not aware of any studies that are comparable to the present paper, both in terms of evaluating the impacts of mixtures of substrates and in modelling their kinetics.

And, finally, with respect to the inactivation mechanisms for laccase, we have mentioned
inactivation in the introduction of the paper (see paragraph beginning with “Mixture effects will likely depend on…”) with appropriate references to prior work that go into further detail that do not warrant being repeated in this paper. Also, the modelling of inactivation, as reflected by Equation (4), is explained in several cited references and particularly in Appendix A (see Section A.1), where we state: “For these three substrates, including estradiol, cumylphenol and triclosan, it was necessary to incorporate an expression accounting for enzyme inactivation into the model [6], where it was assumed that inactivation results from the interaction of substrate radicals produced during the reaction, S∙, with the active enzyme.” As such, we think that we have added enough information to explain the basis of the model and to refer the reader to other sources of information
regarding inactivation.

Round 2

Reviewer 4 Report

The authors were kindly anwered to all comments. I can understand that they have specific preferences about how a manuscript should be written and I have not any doubt about this. My opinion, regarding these preferences, just reflects how a reader "see" your manuscript. Although, the authors didn't make significant changes, this do not affect the overall scientific quality of the manuscript. 

Concerning the enrichment of the discussion, my comment is that the results are not adequately compared with previous studies. Indeed, the assumption of authors, that there not many other comparable studies, is correct. In this case, the authors could state that there not published studies relevant to this subject and highlight by this way the novelty. This makes it easier for a reader to understand the significance and novelty of your study.

The manuscript could be accepted in the present form.